# Cytosolic calcium regulates cytoplasmic accumulation of TDP-43 through Calpain-A and Importin α3

Jeong Hyang Park[1,2†], Chang Geon Chung[1,2†], Sung Soon Park[1,2], Davin Lee[1,2], Kyung Min Kim[1,3], Yeonjin Jeong[1,2], Eun Seon Kim[1,4], Jae Ho Cho[1,2], Yu-Mi Jeon[4], C-K James Shen[5], Hyung-Jun Kim[4], Daehee Hwang[3*], Sung Bae Lee[1,2,4*]

[1]Department of Brain & Cognitive Sciences, DGIST, Daegu, Republic of Korea; [2]Protein dynamics-based proteotoxicity control laboratory, Basic research lab, DGIST, Daegu, Republic of Korea; [3]School of Biological Sciences, Seoul National University, Seoul, Republic of Korea; [4]Dementia research group, Korea Brain Research Institute (KBRI), Daegu, Republic of Korea; [5]Taipei Medical University/ Institute of Molecular Biology, Academia Sinica, Taipei, Taiwan

*For correspondence:
daehee@snu.ac.kr (DH);
sblee@dgist.ac.kr (SBL)

[†]These authors contributed equally to this work

Competing interests: The authors declare that no competing interests exist.

**Abstract** Cytoplasmic accumulation of TDP-43 in motor neurons is the most prominent pathological feature in amyotrophic lateral sclerosis (ALS). A feedback cycle between nucleocytoplasmic transport (NCT) defect and TDP-43 aggregation was shown to contribute to accumulation of TDP-43 in the cytoplasm. However, little is known about cellular factors that can control the activity of NCT, thereby affecting TDP-43 accumulation in the cytoplasm. Here, we identified via FRAP and optogenetics cytosolic calcium as a key cellular factor controlling NCT of TDP-43. Dynamic and reversible changes in TDP-43 localization were observed in *Drosophila* sensory neurons during development. Genetic and immunohistochemical analyses identified the cytosolic calcium-Calpain-A-Importin α3 pathway as a regulatory mechanism underlying NCT of TDP-43. In *C9orf72* ALS fly models, upregulation of the pathway activity by increasing cytosolic calcium reduced cytoplasmic accumulation of TDP-43 and mitigated behavioral defects. Together, these results suggest the calcium-Calpain-A-Importin α3 pathway as a potential therapeutic target of ALS.

## Introduction

Amyotrophic lateral sclerosis (ALS) is a fatal neurodegenerative disease that mainly affects both upper and lower motor neurons, accompanied by motor symptoms (*Taylor et al., 2016*). Cytoplasmic accumulation of TDP-43 in motor neurons is the most prominent pathological feature in ALS, which results in impaired protein quality control, mitochondrial dysfunction, and altered stress granule dynamics (*Shahheydari et al., 2017*; *Davis et al., 2018*; *McDonald et al., 2011*). This cytoplasmic accumulation often accompanies depletion of TDP-43 in the nucleus (*Neumann et al., 2006*), which then results in the loss of its nuclear functions including mRNA splicing (*Highley et al., 2014*) and the regulation of transcription (*Amlie-Wolf et al., 2015*). In addition to its occurrence in ALS, the cytoplasmic accumulation of TDP-43 in neurons is also observed in frontotemporal dementia (FTD; about 50% of patients) (*Neumann et al., 2006*), as well as in Alzheimer's (*Amador-Ortiz et al., 2007*), Parkinson's (*Nakashima-Yasuda et al., 2007*), and Huntington's diseases (*Schwab et al., 2008*), although with lower frequency compared to ALS. These data suggest the importance of proper nucleocytoplasmic localization of TDP-43 for neuronal function.

TDP-43 is known to localize mainly in the nucleus of most neurons examined in non-pathological conditions (*Diaper et al., 2013*; *Neumann et al., 2006*; *Uchino et al., 2015*). Post-translational

modifications (PTMs), such as ubiquitination and phosphorylation, and fragmentation of TDP-43 were shown to affect interactions among TDP-43 protein themselves and between TDP-43 and Importins, thereby promoting cytoplasmic accumulation of TDP-43 under pathological conditions including ALS (*Neumann et al., 2006*; *Nonaka et al., 2009*). Consistent with these findings, recently, a feedback cycle between nucleocytoplasmic transport (NCT) defect and TDP-43 aggregation was proposed as a model to explain cytoplasmic accumulation of TDP-43 in ALS conditions (*Solomon et al., 2018*; *Gasset-Rosa et al., 2019*). In this model, aggregation of TDP-43 in the cytoplasm captures the components (e.g., Importins) of NCT, causing NCT defects and reduced translocation of TDP-43 to the nucleus. These defects increase the amount of TDP-43 in the cytoplasm, which further accelerates cytoplasmic aggregation of TDP-43. Although this model explains the late stages of the disease very well, it remains unclear how this feedback cycle is initiated at the early stages or what predisposes neurons to be vulnerable to this feedback cycle.

Of note, TDP-43 has a property of shuttling between the nucleus and the cytoplasm in neurons under non-pathological conditions, and several factors have been shown to contribute to its NCT. Numerous studies have demonstrated contributions of nuclear transport machineries to NCT of TDP-43, including Importins (Importin α2, α3, and β1) (*Nishimura et al., 2010*; *Solomon et al., 2018*), RanGAP (*Chou et al., 2018*), and components of nuclear pore complexes (Nup62 and Nup54) (*Nishimura et al., 2010*). The signal sequences for NCT of TDP-43 have been reported: the nuclear localization signal (NLS) of TDP-43 was first identified in both cultured neurons and mouse brains, and its deletion resulted in cytoplasmic accumulation of TDP-43 (*Winton et al., 2008*). In contrast, deletion of a computationally predicted, leucine-rich nuclear export signal (NES) located at amino acids 239–250 in human TDP-43 (*Winton et al., 2008*) showed no significant effects on translocation of TDP-43 towards the cytoplasm in neurons (*Pinarbasi et al., 2018*).

Given that TDP-43 has distinct cytoplasmic functions in addition to its nuclear functions, TDP-43 localization can be dynamically changed (i.e. from nucleus to cytoplasm and vice versa) upon specific cellular demands for its location-specific functions. Although some specific pathological conditions, such as increased stress granule formation (*Colombrita et al., 2009*) or decreased autophagy (*Nguyen et al., 2019*), are known to induce cytoplasmic accumulation of TDP-43, it remains elusive whether there exist in neurons certain non-pathological cellular events in which the localization of TDP-43 shifts significantly from the nucleus to cytoplasm and vice versa. In this study, we found that TDP-43 shows cell-type-dependent variation in its localization and can undergo dynamic and reversible changes in its localization during development, one of the major cellular events, in *Drosophila* sensory neurons. Furthermore, we identify cytosolic calcium as a key cellular factor that controls nucleocytoplasmic localization of TDP-43 and establish the cytosolic calcium-Calpain-A-Importin α3 pathway as a molecular mechanism underlying NCT-mediated cytoplasmic accumulation of TDP-43 in *Drosophila* neurons.

## Results

### Proportions of nucleus- and cytoplasm-localized TBPH vary with cell type and developmental stage in *Drosophila* neurons

Previously, TDP-43 has been reported to be localized mainly to the nucleus in various neurons such as motor neurons in *Drosophila* (*Diaper et al., 2013*), mice (*Neumann et al., 2006*), and humans (*Uchino et al., 2015*). Recently, however, around 10–20% of neurons in pontine nuclei, thalamus, CA3 region of hippocampus, and orbital cortex showed nuclear depletion and cytoplasmic accumulation of TDP-43 in mice (*Termsarasab et al., 2020*). Furthermore, it was shown that TDP-43 localized predominantly in the cytoplasm in *Drosophila* larval class IV dendritic arborization (C4da) sensory neurons (*Rumpf et al., 2014*) and differentiating mouse myoblasts (*Vogler et al., 2018*). These pieces of information from multiple studies collectively suggest a hypothesis that TDP-43 is likely to show cell-type-dependent variation in its localization pattern between the nucleus and cytoplasm. To test this hypothesis, we overexpressed Flag-tagged *Drosophila TDP-43* (*TBPH*) in C4da (*ppk*[1a]-*gal4*, membrane marked by HRP staining), motor (*D42-gal4*, membrane marked by mCD8-RFP), and dopaminergic (DA; *TH-gal4*, membrane marked by CD4-tdTom) neurons, and in glial cells (*repo-gal4*, membrane marked by mCD8-RFP) of larvae at 120 hr after egg laying (AEL) using the specified *gal4s* and membrane markers. We quantitatively measured amounts of TBPH localized in

the cytoplasm and the nucleus and then compared cytoplasm-to-nucleus (Cyt/Nuc) ratios in each cell type. The specificity of anti-Flag antibody against TBPH-Flag was confirmed (*Figure 1—figure supplement 1A*). C4da neurons showed significantly ($p<1.0 \times 10^{-4}$) higher Cyt/Nuc ratios than motor and DA neurons and glial cells by fourfold (*Figure 1A and B*), consistent with our hypothesis.

Recently, it has been reported that TDP-43 is localized primarily to the nucleus in uninjured myofibres, but translocated to the cytoplasm in regenerating myofibres following chemical injury in the mouse model (*Vogler et al., 2018*). This observation suggests a hypothesis that TDP-43 can change its localization during the course of cellular events. We tested this hypothesis in *Drosophila* neurons during development, one of prominent cellular events in flies. To this end, we overexpressed *TBPH* in *Drosophila* C4da neurons and then examined TBPH localization patterns along the development. Interestingly, TBPH changed its localization from the cytoplasm (120 hr AEL) to the nucleus in the pupal stage at 18 hr after puparium formation (APF) (*Figure 1C*). TBPH was localized to both the cytoplasm and the nucleus in an intermediate stage (0 hr APF) between larva and pupa (*Figure 1C*), suggesting that TBPH may begin its nuclear entry at this stage. Furthermore, TBPH was localized to both the cytoplasm and the nucleus in an early (10 day; 10d) adult stage and then re-localized mainly to the nucleus in the late (40 day; 40d) adult stage (*Figure 1C*). Quantitative comparisons revealed that Cyt/Nuc ratios between pairs of consecutive developmental stages (120 hr AEL-0 hr APF, 0-18 hr APF, 18 hr APF-early adult, and early-late adult) were significantly different (*Figure 1D*). To check whether overexpression might affect this development-dependent variation in Cyt/Nuc ratios, we further examined the localization of endogenous TBPH in C4da neurons at 120 hr AEL and 18 hr APF. Endogenous TBPH showed localization patterns similar to—although less dramatic than—overexpressed TBPH (*Figure 1—figure supplement 1B and C*). The endogenous TBPH expression level in C4da neurons between the two stages seemed comparable based on the immunostaining data. Taken together, these data suggest that the localization pattern of TBPH between nucleus and cytoplasm in *Drosophila* neurons varies with cell type and developmental stage.

## Cytosolic calcium mediates nucleocytoplasmic translocation of TBPH/TDP-43

Although PTMs of TDP-43 (*Neumann et al., 2006*; *Nonaka et al., 2009*) and some pathological conditions such as increased stress granule formation (*Colombrita et al., 2009*) or decreased autophagy (*Nguyen et al., 2019*) have been shown to be associated with its nucleocytoplasmic translocation, little is known about cellular factors that govern the nucleocytoplasmic translocation of TDP-43. Along the course of their development, the intracellular calcium levels in neurons are known to undergo dynamic changes; for example, it increases in *Drosophila* C4da neurons during the pupal period for dendrite pruning (*Kanamori et al., 2013*), and calcium influx also increases in rat hippocampal neurons during aging (*Gant et al., 2006*). Together with these previous observations, our data (*Figure 1C*) suggest a hypothesis that intracellular calcium may act as a cellular factor determining the nucleocytoplasmic translocation of TBPH.

To test this hypothesis, we examined the association of Cyt/Nuc ratios of TBPH with calcium levels in C4da neurons at two developmental stages, larva (120 hr AEL) and pupa (18 hr APF). We measured the calcium level (*Figure 2A*) using the intensity ratio of genetically encoded calcium indicator (GCaMP) to red fluorescent protein tdTomato (tdTom) as previously described (*Daniels et al., 2014*). GCaMP/tdTom mean intensity ratio in pupae was significantly increased compared to that in larvae (*Figure 2B*), suggesting that calcium levels have an inverse relationship with Cyt/Nuc ratios of TBPH. We further examined whether manipulation of intracellular calcium level could alter Cyt/Nuc ratios of TBPH. To this end, given the relatively higher intracellular calcium level in pupae, we decreased intracellular calcium level using previously reported *Itpr*$^{ka1091}$ mutants (*Joshi et al., 2004*), which have a decreased activity of Inositol (1,4,5)-trisphosphate receptor (Itpr), a major calcium-releasing channel in the endoplasmic reticulum (ER). In the *Itpr*$^{ka1091/+}$ pupae, a significantly ($p<0.01$) higher amount of TBPH was localized to the cytoplasm compared to the controls, resulting in an increased Cyt/Nuc ratio (*Figure 2C and D*). Similar results were found using another *Itpr* mutant (*Itpr*$^{sv35/+}$) and *Itpr* RNAi (*Figure 2—figure supplement 1A and B*). These results are consistent with the previous observation of increased Cyt/Nuc ratio of human TDP-43 in HeLa and SH-SY5Y cells as well as in primary rat cortical neurons upon knockdown of *ITPR1* (*Kim et al., 2012*). Similarly, mutation (*RyR*$^{16/+}$) of *ryanodine receptor*, another major calcium-releasing channel in the ER, led to a similar increase in the Cyt/Nuc ratio of TBPH (*Figure 2—figure supplement 1C and D*).

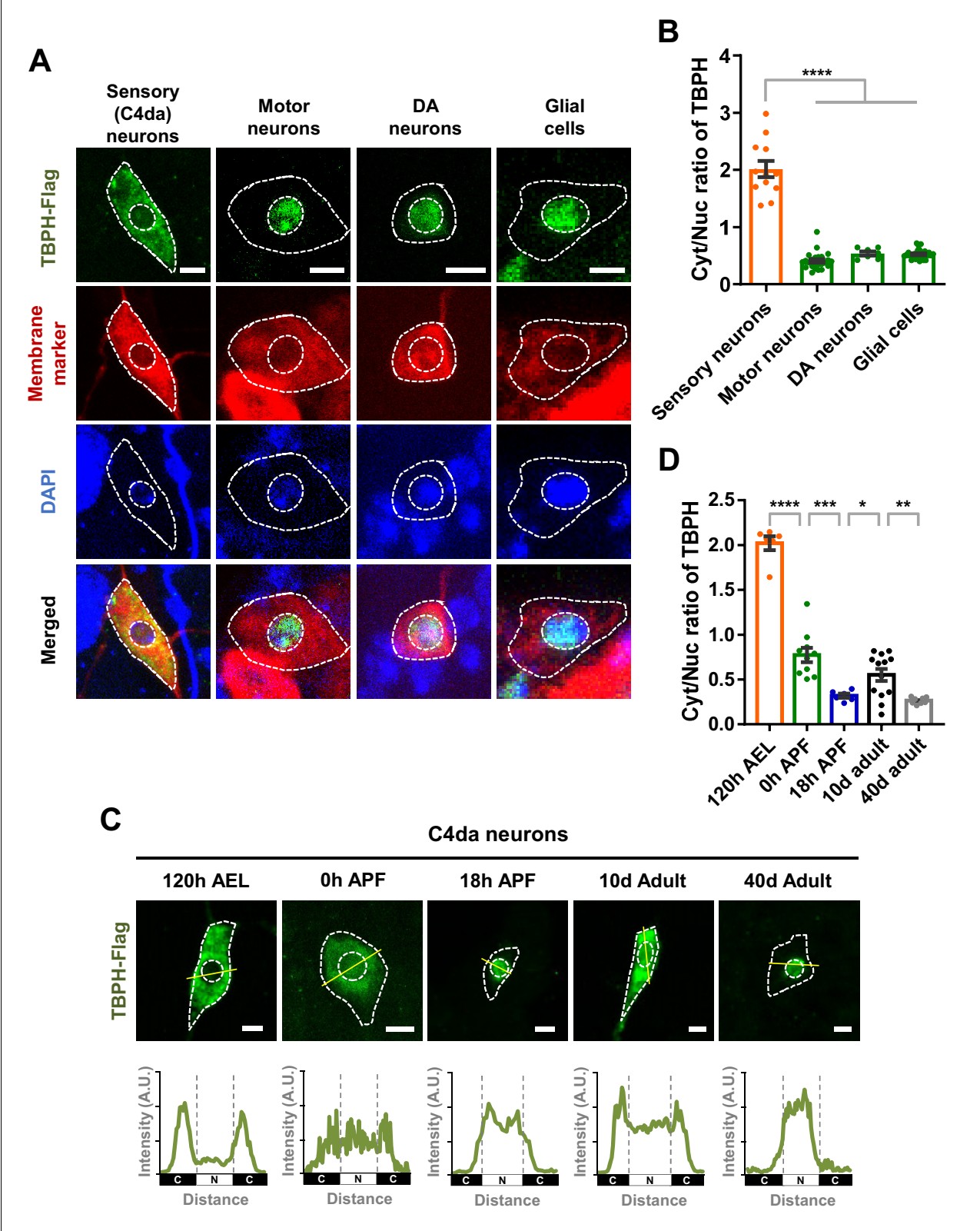

**Figure 1.** Cell type- and developmental stage-dependent variation in nucleocytoplasmic localization of TBPH. (A) Subcellular localization of overexpressed TBPH-Flag proteins in sensory (C4da), motor, and dopaminergic (DA) neurons, and in glial cells [Genotype: Sensory (C4da) neurons, +/+;UAS-TBPH-Flag-HA/ppk[1a]-Gal4, Motor neurons, +/+;UAS-TBPH-Flag-HA/D42-Gal4,UAS-mCD8-RFP, DA neurons, +/+;UAS-TBPH-Flag-HA/TH-Gal4, UAS-CD4-tdTom, Glial cells, +/+;UAS-TBPH-Flag-HA/repo-Gal4,UAS-mCD8-RFP]. DAPI staining was used to mark the nuclei. Merged

*Figure 1 continued on next page*

*Figure 1 continued*

immunohistochemical images of TBPH proteins (green), plasma membranes (red), and DAPI (blue) are presented at the bottom. Outer and inner dashed lines indicate borders of cell bodies and nuclei, respectively (Scale bars, 5 μm). (B) Quantification of cytoplasmic/nuclear (Cyt/Nuc) ratio of TBPH-Flag proteins in four different cell types described in A. ****$p<1.0 \times 10^{-4}$ by one-way ANOVA with Tukey's post-hoc correction; error bars, ± SEM; n = 12 for sensory neurons, n = 24 for motor neurons, n = 7 for DA neurons, n = 28 for glial cells. (C) Subcellular localization of overexpressed TBPH-Flag proteins in C4da neurons at five different developmental time points (120 hr AEL, 0 hr APF, 18 hr APF, 10-day adult, and 40-day adult) [Genotype: *+/+;UAS-TBPH-Flag-HA/ppk$^{1a}$-Gal4,UAS-mCD8-RFP*]. Outer and inner dashed lines indicate the borders of cell bodies and nuclei, respectively (scale bars, 5 μm). The intensity profile of fluorescent signals representing TBPH proteins across cell bodies along yellow lines are presented at the bottom. The gray dashed lines mark borders of nuclei (bottom panels). (D) Quantification of Cyt/Nuc ratio of TBPH-Flag proteins at five different developmental time points described in C. ****$p<1.0 \times 10^{-4}$, ***p=0.0007, **p=0.0042, *p=0.0369 by two-tailed t-test; error bars, ± SEM; n = 6 for 120 hr AEL, n = 9 for 0 hr APF, n = 6 for 18 hr APF, n = 11 for 10-day adult, n = 8 for 40-day adult.

The online version of this article includes the following source data and figure supplement(s) for figure 1:

**Source data 1.** Numerical data plotted in *Figure 1B, D* and *Figure 1—figure supplement 1B*.
**Figure supplement 1.** Developmental stage-dependent changes in nucleocytoplasmic localization of endogenous TBPH.

Given the relatively lower intracellular calcium level in larvae, we next increased the intracellular calcium level through a knockdown (*SERCA Ri*) of the Sarco/endoplasmic reticulum calcium ATPase (*SERCA*), a major calcium uptake pump in the ER. A higher GCaMP/tdTom mean intensity ratio confirmed the increased intracellular calcium level in the *SERCA RNAi* (*Ri*) larva (*Figure 2—figure supplement 1E*). In the *SERCA Ri* larvae, the majority of TBPH was translocated to the nucleus, resulting in a significantly (p<1.0 × $10^{-4}$) decreased Cyt/Nuc ratio (*Figure 2E and F*). We then increased the intracellular calcium level by overexpressing *NachBac* (*NachBac O/E*), a plasma membrane sodium channel, known to increase intracellular calcium level (*Nitabach et al., 2006*). Consistently, *NachBac* overexpression led to a similar decrease in Cyt/Nuc ratio of TBPH to that in the *SERCA Ri* larvae (*Figure 2—figure supplement 1F and G*).

Knockdown and overexpression experiments lead to chronic changes in calcium level, potentially activating compensatory mechanisms that may be responsible for the redistribution of TBPH. To reduce the possibility of activating compensatory mechanism from chronically increasing cytosolic calcium, we decided to increase intracellular calcium level in larval C4da neurons using an optogenetics technique. We first confirmed that upon optogenetic activation the larvae raised on food containing all trans-retinal (ATR) showed a mild increase (~20%) in intracellular calcium level in C4da neurons expressing channelrhodpsin (*Figure 2—figure supplement 1H and I*; *Figure 2—video 1*). This increase in calcium was sufficient to cause rolling response in larvae, a behavior previously shown to be elicited by increased intracellular calcium in C4da neurons (*Figure 2—video 2*; *Hwang et al., 2007*; *Kaneko et al., 2017*). Next, the larvae expressing channelrhodpsin and TBPH-Flag in C4da neurons were raised on food containing ATR. When the larvae reached the wandering third instar stage, they were put under the blue light (470 nm) for 30 min and then in the dark for 1 hr, and this light on/off cycle was repeated four times to a total of 120 min of light exposure. After the last off cycle of 1 hr, the larvae were dissected and immunostained for TBPH using anti-Flag antibody before imaging (*Figure 2G*). The Cyt/Nuc ratio of TBPH significantly (p<0.05) decreased compared to the controls that did not receive any blue light (*Figure 2H and I*). We also expressed RFP-tagged human TDP-43 in C4da neurons and applied the same optogenetics protocol (*Figure 2G*) before imaging them live. A similar decrease in Cyt/Nuc ratio was observed for the RFP-TDP-43 (*Figure 2—figure supplement 1J and K*) upon optogenetic stimulation. When we decreased the total amount of larval exposure to blue light to just 5 min, the Cyt/Nuc ratio of RFP-TDP-43 did not change at any of the time points (1, 30, 60, 90, and 120 min) imaged after the exposure (data not shown), suggesting that 5 min of increased cytosolic calcium is not enough to alter RFP-TDP-43 localization.

Next, we examined the subcellular localization of ALS-linked mutant TDP-43 G287S (*Voigt et al., 2010*) and tested whether developmental process or calcium can alter its localization. As expected, the localization of TDP-43 WT significantly changed from the cytoplasm in larvae to the nucleus in pupae (*Figure 2—figure supplement 2A and B*). The localization of TDP-43 G287S also shifted from the cytoplasm to the nucleus, albeit less so compared to TDP-43 WT. When we decreased cytosolic calcium via *ryanodine receptor* (*RyR*) knockdown in pupae, the Cyt/Nuc ratio of TDP-43 WT increased significantly. The localization of TDP-43 G287S also shifted back toward the cytoplasm,

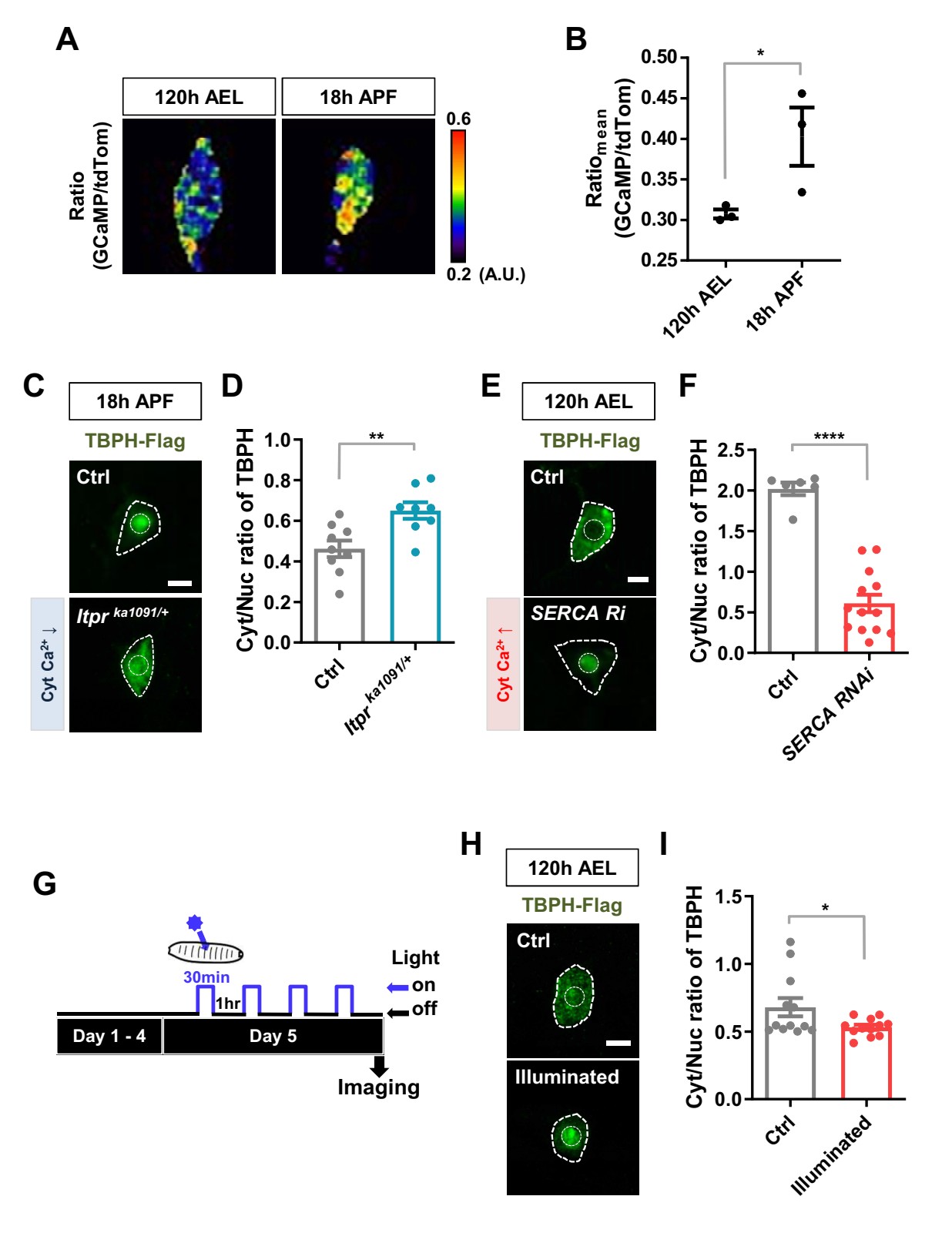

**Figure 2.** Regulation of nucleocytoplasmic translocation of TBPH by cytoplasmic calcium. (**A**) Representative pseudo-colored images representing relative intensity ratios (i.e. calcium level) of GCaMP over tdTom (i.e. overexpressed membrane marker proteins used as a control) in C4da neurons at 120 hr AEL and 18 hr APF [Genotype: *+/+;UAS-tdTomato P2A GCaMP5G/ppk[1a]-Gal4*]. (**B**) Quantification of GCaMP/tdTom mean intensity ratios at 120 hr AEL and 18 hr APF. *p=0.0295 by one-tailed t-test; error bars,± SEM; n = 3 neurons. (**C**) Subcellular localization of overexpressed TBPH-Flag proteins

*Figure 2 continued on next page*

*Figure 2 continued*

in C4da neurons of Ctrl or *Itpr*[ka1091/+] mutants (*Itpr*[ka1091/+]) at 18 hr APF [Genotype: Ctrl, *+/+;ppk*[1a]*-Gal4,UAS-TBPH-Flag-HA/+, Itpr*[ka1091/+], *+/+;ppk*[1a]*-Gal4,UAS-TBPH-Flag-HA/Itpr*[ka1091]]. Outer and inner dashed lines indicate borders of cell bodies and nuclei, respectively (Scale bar, 5 μm). (D) Quantification of Cyt/Nuc ratio of TBPH-Flag proteins in C4da neurons of Ctrl or *Itpr*[ka1091/+] at 18 hr APF. **p=0.0054 by two-tailed t-test; error bars,± SEM; n = 9 for Ctrl, n = 8 for *Itpr*[ka1091/+]. (E) Subcellular localization of overexpressed TBPH-Flag proteins in C4da neurons of Ctrl or expressing SERCA RNAi (*SERCA Ri*) at 120 hr AEL [Genotype: Ctrl, *+/+;ppk*[1a]*-Gal4,UAS-TBPH-Flag-HA/+, SERCA Ri, UAS-SERCA RNAi/+;ppk*[1a]*-Gal4,UAS-TBPH-Flag-HA/+*]. Outer and inner dashed lines indicate borders of cell bodies and nuclei, respectively (Scale bar, 5 μm). (F) Quantification of Cyt/Nuc ratio of TBPH-Flag proteins in C4da neurons of Ctrl or expressing *SERCA Ri* at 120 hr AEL. ****$p<1.0\times10^{-4}$ by two-tailed t-test; error bars,± SEM; n = 6 for Ctrl, n = 13 for *SERCA Ri*. (G) Experimental scheme of optogenetics. The blue light (470 nm) was applied four times to the larvae at 5 days AEL to optogenetically stimulate C4da neurons expressing channelrhodopsin. (H) Subcellular localization of overexpressed TBPH-Flag proteins in C4da neurons of Ctrl (not illuminated) or illuminated larvae [Genotype: *20XUAS-Chr2.T159C-HA/+;UAS-ChR2.S/ppk*[1a]*-Gal4,UAS-TBPH-Flag-HA*]. Outer and inner dashed lines indicate borders of cell bodies and nuclei, respectively (scale bar, 5 μm). (I) Quantification of Cyt/Nuc ratio of TBPH-Flag proteins in C4da neurons of Ctrl (not illuminated) or illuminated larvae. *p=0.0447 by two-tailed t-test; error bars, ± SEM; n = 12 neurons.

The online version of this article includes the following video, source data, and figure supplement(s) for figure 2:

**Source data 1.** Numerical data plotted in *Figure 2b, D, F, I* and *Figure 2—figure supplement 1B, D, G, I, J* and *2B*.

**Figure supplement 1.** Regulation of nucleocytoplasmic translocation of TBPH/TDP-43 by cytoplasmic calcium.

**Figure supplement 2.** Developmentally regulated nucleocytoplasmic translocation of TDP-43 and ALS-linked TDP-43 G287S modulated by cytoplasmic calcium.

**Figure 2—video 1.** Rolling response of larva upon optogenetic stimulation of C4da neurons expressing channelrhodopsin [Genotype: *20XUAS-ChR2.T159C-HA/+;UAS-ChR2.S/ppk*[1a]*-Gal4*].
https://elifesciences.org/articles/60132#fig2video1

**Figure 2—video 2.** Monitoring dynamic changes of intracellular calcium level in a C4da neuron before and after optogenetic stimulation.
https://elifesciences.org/articles/60132#fig2video2

but the magnitude of the shift was again smaller than that of TDP-43 WT (*Figure 2—figure supplement 2A and B*). These data suggest that TDP-43 G287S is less sensitive than TDP-43 WT to the changes in development and cytosolic calcium in mediating NCT. Overall, altering cytosolic calcium in a variety of ways led to significant changes in the localization of TBPH/TDP-43, suggesting that calcium—and not other secondary causes arising from altering calcium-associated channels—regulates the nucleocytoplasmic localization of TBPH/TDP-43.

From these experiments, it is difficult to tell whether there is a decrease in nuclear export or an increase in nuclear import of TDP-43 when cytosolic calcium is increased. To test which is the case, we employed fluorescence recovery after photobleaching (FRAP) and optogenetic techniques to characterize the mobility of RFP-TDP-43 across the nuclear envelope upon stimulation (*Figure 3A*). First, we raised one group of larvae in food with ATR (ATR+) and another in food without ATR (ATR-). We then optogenetically stimulated third instar larvae from both groups for 30 min, increasing the intracellular calcium level only in those raised in ATR+. After incubation in the dark for 2 hr, we photobleached the nucleus of C4da neurons of larvae from both groups and examined the fluorescence recovery of RFP-TDP-43 (*Figure 3B*). C4da neurons in larvae that were raised in ATR+ food showed a significantly faster ($t_{1/2}$: 24.9 s; $t_{plateau}$: 146.3 s; $slope_{max}$: 0.01498) recovery of fluorescence in the nucleus compared to those raised in ATR- ($t_{1/2}$: 46.7 s; $t_{plateau}$: 200.9 s; $slope_{max}$: 0.00528) (*Figure 3C and D*). Taken together, these data suggest that cytosolic calcium facilitates nuclear import of TBPH in C4da neurons in larvae and pupae.

## Calcium-dependent regulators and nuclear import components regulate nucleocytoplasmic translocation of TBPH/TDP-43

Next, we asked how cytosolic calcium mediates nuclear import of TBPH. TBPH itself has no calcium-binding motif, suggesting a possibility that calcium indirectly mediates its nucleocytoplasmic translocation through calcium-dependent regulators. To test this possibility, we performed an RNAi screen of 12 calcium-dependent regulators (*Figure 4A*) using pupal C4da neurons. Considering that TBPH is primarily localized to the nucleus in pupae, we searched for calcium-dependent regulators that could shift the TBPH localization toward the cytoplasm when their expression was reduced. Among the 12 regulators, knockdown of *Calmodulin* (*Cam*), *Protein kinase C* (*Pkc53E*), and *Calpain-A* (*CalpA*) considerably shifted the localization of TBPH toward the cytoplasm compared to the controls (*Figure 4B*), thereby significantly (p<0.05) increasing Cyt/Nuc ratios (*Figure 4C*).

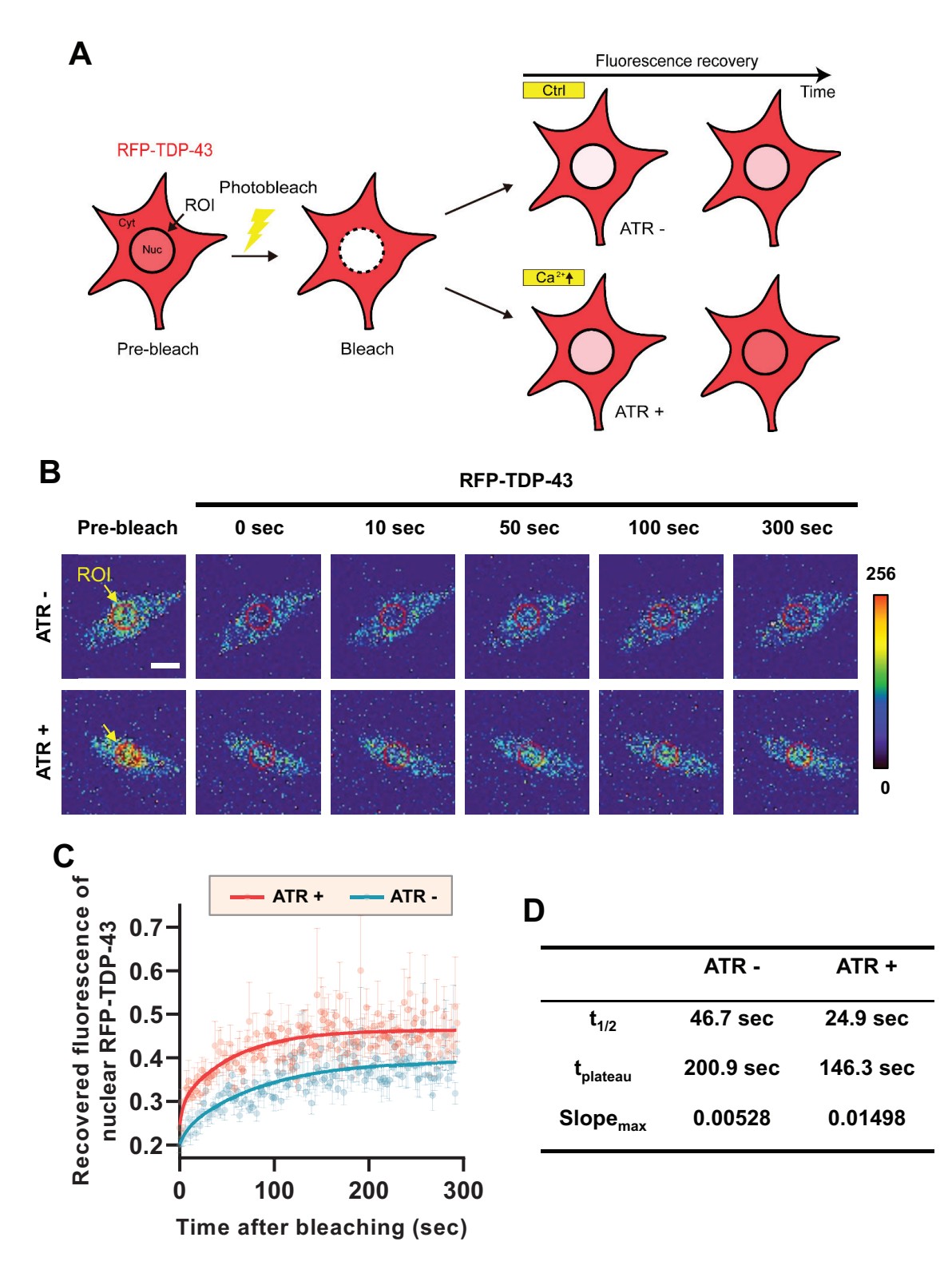

**Figure 3.** Optogenetic stimulation facilitates nuclear import of RFP-TDP-43 in C4da neurons. (A) An experimental scheme of fluorescence recovery after photobleaching (FRAP) analysis of RFP-TDP-43 in the nucleus to assess its nuclear import after optogenetic stimulation. (B) Time-lapsed images of RFP-TDP-43 (pseudo-colored image) nuclear import dynamics in C4da neurons. Nuclear area is selected as region of interest (ROI) (scale bar, 5 μm). (C) FRAP analysis comparing recovery kinetics of RFP-TDP-43 signal in the nucleus of C4da neurons of larvae raised in food with no ATR (ATR -) or with ATR

*Figure 3 continued on next page*

Figure 3 continued

(ATR +) at 120 hr AEL [Genotype: *20XUAS-ChR2.T159C-HA/+;UAS-3xMyc-RFP-TDP-43/ppk[1a]-Gal4*]. error bars,± SEM; n = 5 for ATR-, n = 7 for ATR+.
(D) Analysis of FRAP data from C.
The online version of this article includes the following source data for figure 3:

**Source data 1.** Numerical data plotted in *Figure 3C*.

The NLS of TBPH has been well-characterized (*Winton et al., 2008*) and we have shown above that upon optogenetic stimulation (*Figure 2H and I*), TBPH is actively transported into the nucleus. In contrast, the NES of TBPH is poorly understood, and the NES of TDP-43 was shown to be insufficient to export TDP-43 from the nucleus (*Pinarbasi et al., 2018*). Therefore, we performed an RNAi screen of 12 components involved in nuclear import (*Figure 4—figure supplement 1A*) using pupal C4da neurons. Among the 12 components, knockdown of *Importin alpha 1* (*Imp α1*), *Importin alpha 3* (*Imp α3*), *Importin beta 1* (*Imp β1*), *Importin 7* (*Imp 7*), and *Transportin-Serine/Arginine rich* (*Tnpo-SR*) considerably shifted the localization of TBPH toward the cytoplasm (*Figure 4—figure supplement 1B*) and significantly increased Cyt/Nuc ratios (*Figure 4—figure supplement 1C*). We then overexpressed *Imp α3* and *Imp β1* (knockdown of which resulted in the largest increase in Cyt/Nuc ratio) in larval C4da neurons and examined the changes in Cyt/Nuc ratio of TBPH. Consistent with the findings from knockdown experiments, overexpression of *Imp α3* (*Imp α3 O/E*), *Imp β1* (*Imp β1 O/E*), or *Imp α3* and *Imp β1* (*Imp α3 O/E + Imp β1 O/E*) shifted the localization of TBPH from the cytoplasm to the nucleus (*Figure 4—figure supplement 2A and B*). Taken together, these data suggest that calcium-dependent regulators and nuclear import components control nucleocytoplasmic translocation of TBPH.

## Calpain-A controls the nucleocytoplasmic distribution of Importin α3

The above data suggest a possible regulatory relationship between calcium-dependent regulators and nuclear import components. To test this regulatory relationship, we chose two representative molecules, CalpA and Imp α3, from the selected calcium-dependent regulators and nuclear import components, respectively. According to the genetic screen, knockdown of *Imp α3* (*Figure 4—figure supplement 1C*) and *CalpA* (*Figure 4C*) showed the largest and second largest effect on Cyt/Nuc ratio of TBPH, respectively. Although knockdown of *Cam* showed the largest effect on Cyt/Nuc ratio of TBPH (*Figure 4C*), we focused more on *CalpA* rather than *Cam* for further mechanistic study because co-overexpression of *TBPH* with *Cam* in C4da neurons induced lethality. We next generated transgenic fly lines co-overexpressing flag-tagged *Imp α3* and myc-tagged *CalpA* in larval C4da neurons and examined the localization pattern of Imp α3. In control larvae, Imp α3 localized mainly in the cytoplasm (*Figure 5—figure supplement 1A and B*). Co-overexpression of *CalpA*, however, increased the amount of Imp α3 in the nucleus compared to the controls, and significantly (p<1.0 × $10^{-3}$) decreased the Cyt/Nuc ratio of Imp α3 (*Figure 5—figure supplement 1A and B*). In control pupae, Imp α3 is localized mainly in the nucleus. Knockdown of *CalpA* significantly shifted the localization of Imp α3 toward the cytoplasm compared to the controls, thereby increasing the Cyt/Nuc ratio of Imp α3 (*Figure 5A and B*). Knockdown of *SERCA*, which increases the amount of cytosolic calcium, also significantly (p<0.05) decreased Cyt/Nuc ratio of Imp α3 (*Figure 5C and D*), similar to *CalpA* co-overexpression. These data suggest that both cytosolic calcium and CalpA positively regulate nuclear localization of Imp α3.

We next questioned whether cytosolic calcium induces nuclear localization of Imp α3 via CalpA. To answer this question, we examined the nucleocytoplasmic localization of Imp α3 in larval C4da neurons co-overexpressing *SERCA Ri* and *CalpA Ri* (*SERCA Ri + CalpA Ri*). In these C4da neurons, Imp α3 was predominantly localized in the cytoplasm, implying that the effect of *SERCA* knockdown was abolished by the *CalpA* knockdown (*Figure 5C and D*). We showed above that TBPH was predominantly localized to the nucleus in larval C4da neurons overexpressing *SERCA Ri*. However, co-overexpression of *SERCA Ri* and *Imp α3 Ri* (*SERCA Ri + Imp α3 Ri*) reversed the TBPH localization from the nucleus to the cytoplasm (*Figure 5E and F*). These data collectively suggest that cytosolic calcium controls Imp α3-mediated nuclear localization of TBPH via CalpA (*Figure 5G*).

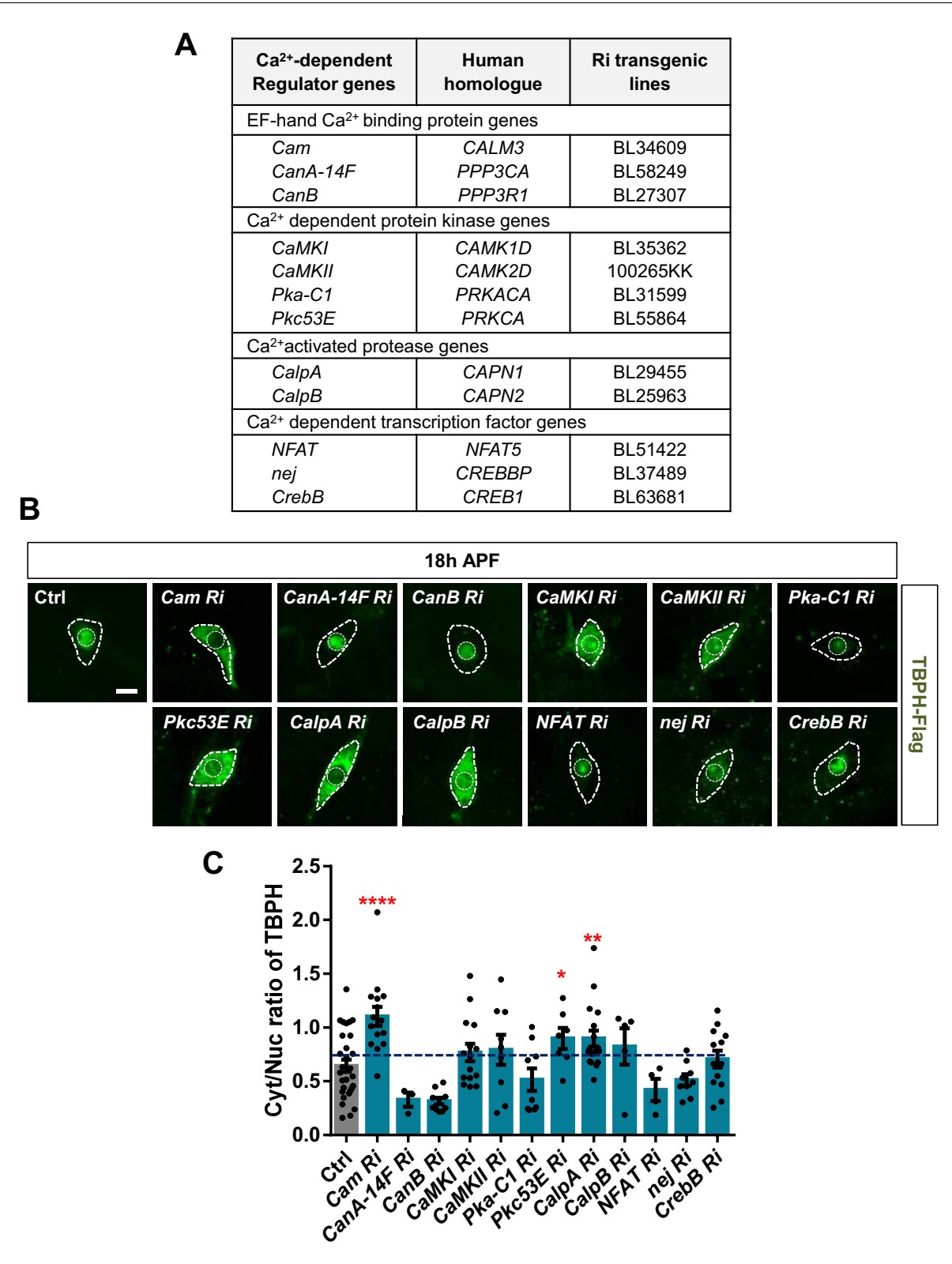

**Figure 4.** Identification of calcium-dependent regulators associated with nucleocytoplasmic translocation of TBPH by a genetic screen. (A) List of calcium-dependent regulators screened in this study. (B) Subcellular localization of overexpressed TBPH-Flag proteins co-overexpressed with denoted RNAi (Ri) transgenes in C4da neurons at 18 hr APF [Genotype: Ctrl, +/+;ppk[1a]-Gal4,UAS-TBPH-Flag-HA/+, Cam Ri, +/+;ppk[1a]-Gal4,UAS-TBPH-Flag-HA/UAS-Cam RNAi, CanA-14F Ri, UAS-CanA-14F RNAi/+;ppk[1a]-Gal4,UAS-TBPH-Flag-HA/+, CanB Ri, +/+;ppk[1a]-Gal4,UAS-TBPH-Flag-HA/UAS-CanB RNAi, *Figure 4 continued on next page*

*Figure 4 continued*

*CaMKI Ri, +/+;ppk$^{1a}$-Gal4,UAS-TBPH-Flag-HA/UAS-CaMKI RNAi/+, CaMKII Ri, UAS-CaMKII RNAi/+;ppk$^{1a}$-Gal4,UAS-TBPH-Flag-HA/+, Pka-C1 Ri, +/+; ppk$^{1a}$-Gal4,UAS-TBPH-Flag-HA/UAS-Pka-C1 RNAi, Pkc53E Ri, +/+;ppk$^{1a}$-Gal4,UAS-TBPH-Flag-HA/UAS-Pkc53E RNAi, CalpA Ri, +/+;ppk$^{1a}$-Gal4,UAS-TBPH-Flag-HA/UAS-CalpA RNAi, CalpB Ri, +/+;ppk$^{1a}$-Gal4,UAS-TBPH-Flag-HA/UAS-CalpB RNAi, NFAT Ri, +/+;ppk$^{1a}$-Gal4,UAS-TBPH-Flag-HA/UAS-NFAT RNAi, nej Ri, +/+;ppk$^{1a}$-Gal4,UAS-TBPH-Flag-HA/UAS-nej RNAi, CrebB Ri, UAS-CrebB RNAi/+;ppk$^{1a}$-Gal4,UAS-TBPH-Flag-HA/+].* Outer and inner dashed lines indicate borders of cell bodies and nuclei, respectively (scale bar, 5 µm). (**C**) Quantification of Cyt/Nuc ratio of TBPH-Flag proteins in C4da neurons expressing denoted transgenes described in B. ****p<1.0×10$^{-4}$, **p=0.0053, *p=0.0276 by one-tailed t-test; error bars, ± SEM; n = 29 for Ctrl, n = 15 for *Cam Ri*, n = 3 for *CanA-14F Ri*, n = 10 for *CanB Ri*, n = 15 for *CaMKI Ri*, n = 9 for *CaMKII Ri*, n = 9 for *Pka-C1 Ri*, n = 7 for *Pkc53E Ri*, n = 17 for *CalpA Ri*, n = 5 for *CalpB Ri*, n = 4 for *NFAT Ri*, n = 9 for *nej Ri*, n = 13 for *CrebB Ri*.

The online version of this article includes the following source data and figure supplement(s) for figure 4:

**Source data 1.** Numerical data plotted in *Figure 4C* and *Figure 4—figure supplement 1C* and *2B*.
**Figure supplement 1.** Identification of nuclear import components associated with nucleocytoplasmic translocation of TBPH by a genetic screen.
**Figure supplement 2.** Regulation of TBPH nuclear import by Imp α3 and Imp β1.

## An untimely nuclear mis-localization of TDP-43 in larval C4da neurons is associated with dysregulation of dendrite arborization

We have shown above that changes in the cytosolic calcium level can markedly shift the localization of TBPH from the cytoplasm to the nucleus and vice versa. This suggests a possibility that upon specific cellular demands for its location-specific functions, TBPH localization might be dynamically changed. Moreover, in cells with lower cytosolic calcium level, such as in larval C4da neurons, cellular demands for nuclear function of TBPH may be minimal. If this is true, inducing nuclear localization of TBPH or TDP-43 in larval C4da neurons may lead to an untimely function and thus an undesirable outcome. To test this, we expressed in larval C4da neurons *TDP-43* (*TDP-43 WT O/E*) or *TDP-43* with mutations to NLS (*TDP-43-ΔNLS O/E*) and examined neuronal morphology as a readout for cell biological consequence of nuclear TDP-43 localization (*Figure 6A*). Previous studies showed that the expression of *TDP-43* can alter dendrite morphology (*Lu et al., 2009*; *Schwenk et al., 2016*; *Herzog et al., 2017*; *Herzog et al., 2020*), but the extent to which its nuclear function is involved remains less known. Interestingly, we found that C4da neurons expressing *TDP-43* showed more aberrant dendrite arborization pattern compared to those expressing *TDP-43-ΔNLS* (*Figure 6A*). Although *TDP-43-ΔNLS* has been shown to be more toxic in some model systems such as rat cortical neurons (*Barmada et al., 2010*), a study using the same constructs as we did reported stronger toxicity induced by wild-type form of *TDP-43* than by *TDP-43-ΔNLS* in certain types of cells (*Miguel et al., 2011*). Further analysis showed that C4da neurons expressing *TDP-43* exhibited significantly shorter total dendritic length (*Figure 6B*) and fewer number of dendritic branch points compared to those expressing *TDP-43-ΔNLS* (*Figure 6C*). Also, sholl analysis indicated that the complexity of dendritic arbor in C4da neurons expressing *TDP-43* was significantly reduced compared to those expressing *TDP-43-ΔNLS* and controls (*Figure 6D*). Consistently, manipulating cytosolic calcium to alter the localization of overexpressed TBPH also led to similar dendritic phenotypes. *SERCA* knockdown, which promotes nuclear translocation of overexpressed TBPH, decreased the total dendritic length (*Figure 6E and F*) and the number of dendritic branch points (*Figure 6E and G*) in larval C4da neurons. *Itpr* knockdown (*Figure 6E and G*) or mutants (*Figure 6—figure supplement 1A–1C*), which inhibit nuclear translocation of overexpressed TBPH, increased the total dendritic length and the number of dendritic branch points. Moreover, overexpression and knockdown of *Imp α3* significantly decreased and increased, respectively, the total dendritic lengths (*Figure 6—figure supplement 1D and E*) and the number of dendritic branch points (*Figure 6—figure supplement 1D and F*) of C4da neurons overexpressing *TBPH*. Furthermore, *Imp α3 Ri* significantly increased the dendritic complexity of *TBPH O/E*, whereas *Imp α3 O/E* decreased the complexity (*Figure 6—figure supplement 1G*). These data indicate that increasing the dosage of nuclear—but not cytoplasmic—TBPH is highly toxic in larval C4da neurons.

To examine through which molecular complex nuclear TBPH induces aberrant dendrite arborization phenotype, we screened for splicing factors (*Figure 6—figure supplement 2A*) whose knockdown mitigated nuclear *TBPH*-mediated dendrite phenotype. We identified through the screen *Hrb27C* (*Figure 6—figure supplement 2B–2E*), the human homolog for which is *DAZAP1* (*Matunis et al., 1992*). Taken together, these results support the notion that expression of TBPH/TDP-43 in a location different from its original cell-type-specific location is deleterious, whether it is

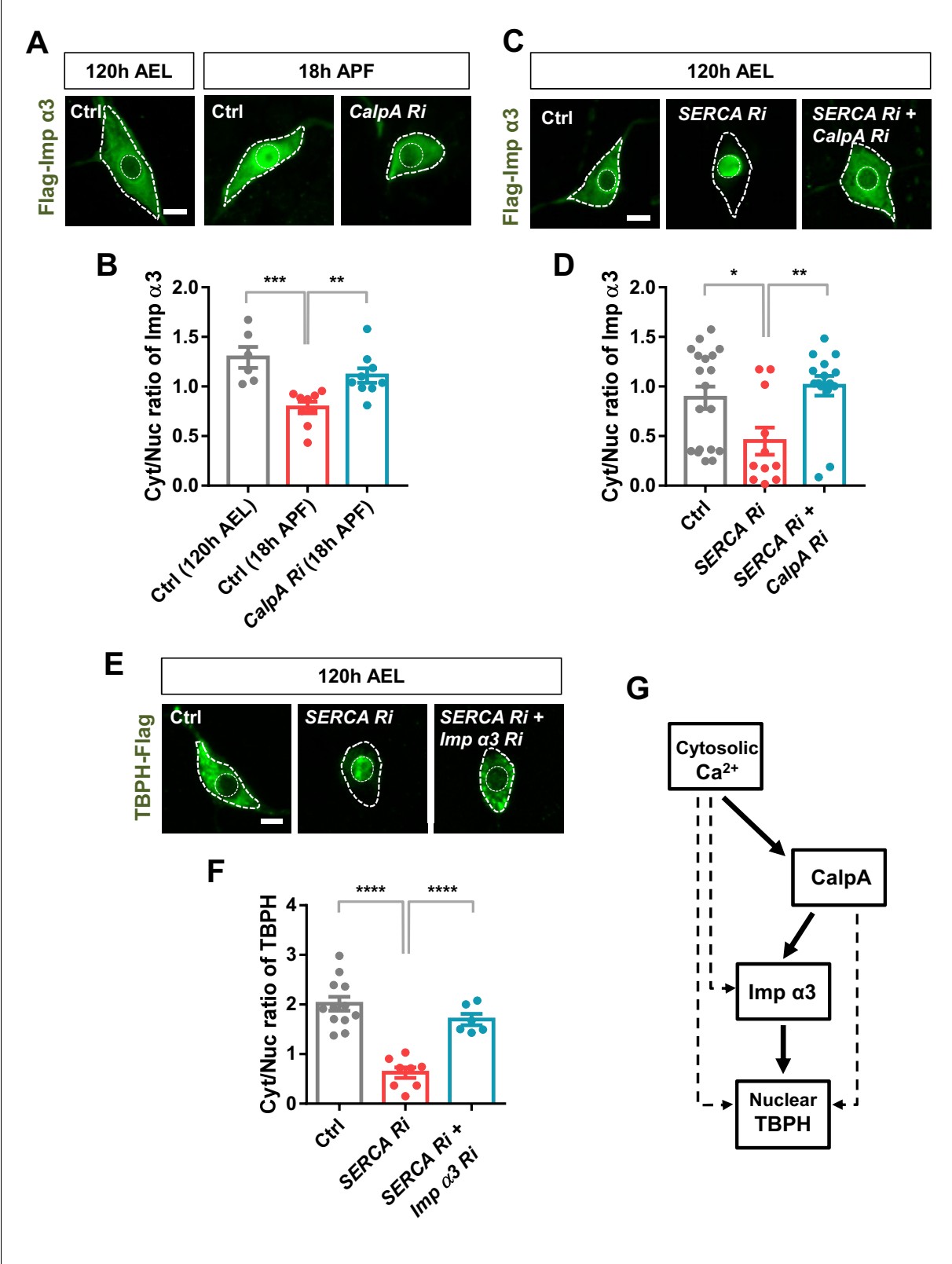

**Figure 5.** Regulation of nucleocytoplasmic translocation of TBPH by the calcium-CalpA-Imp α3 pathway. (**A**) Subcellular localization of overexpressed Flag-Imp α3 proteins in C4da neurons of Ctrl (at 120 hr AEL and 18 hr APF) or with *CalpA* knockdown (*CalpA Ri*) at 18 hr APF [Genotype: Ctrl, *UAS-2xFlag-Imp α3/+;ppk^{1a}-Gal4/+, CalpA Ri, UAS-2xFlag-Imp α3;ppk^{1a}-Gal4/UAS-CalpA RNAi*]. Outer and inner dashed lines indicate borders of cell bodies and nuclei, respectively (scale bar, 5 µm). (**B**) Quantification of Cyt/Nuc ratio of Flag-Imp α3 proteins in C4da neurons of Ctrl (at 120 hr AEL and
*Figure 5 continued on next page*

*Figure 5 continued*

18 hr APF) or with *CalpA* knockdown (*CalpA Ri*) at 18 hr APF. ***p=0.0005, **p=0.0031 by two-tailed t-test; error bars, ± SEM; n = 6 for Ctrl of 120 hr AEL, n = 9 for Ctrl and *CalpA Ri* of 18 hr APF. (**C**) Subcellular localization of overexpressed Flag-Imp α3 proteins in C4da neurons of Ctrl or expressing *SERCA Ri* or both *SERCA Ri* and *CalpA Ri* at 120 hr AEL [Genotype: Ctrl, *UAS-2xFlag-Imp α3/+;ppk^{1a}-Gal4/+*, SERCA Ri, *UAS-2xFlag-Imp α3/UAS-SERCA RNAi;ppk^{1a}-Gal4/+*, SERCA Ri + CalpA Ri, *UAS-2xFlag-Imp α3/UAS-SERCA RNAi;ppk^{1a}-Gal4/UAS-CalpA RNAi*]. Outer and inner dashed lines indicate borders of cell bodies and nuclei, respectively (scale bar, 5 µm). (**D**) Quantification of Cyt/Nuc ratio of Flag-Imp α3 proteins in C4da neurons of Ctrl or expressing *SERCA Ri* or both *SERCA Ri* and *CalpA Ri* at 120 hr AEL. **p=0.0024, *p=0.0219 by two-tailed t-test; error bars, ± SEM; n = 19 for Ctrl, n = 11 for *SERCA Ri*, n = 15 for *SERCA Ri + CalpA Ri*. (**E**) Subcellular localization of overexpressed TBPH-Flag proteins in C4da neurons of Ctrl or expressing *SERCA Ri* or both *SERCA Ri* and *Imp α3 Ri* at 120 hr AEL [Genotype: Ctrl, *+/+;ppk^{1a}-Gal4,UAS-TBPH-Flag-HA/+*, SERCA Ri, *UAS-SERCA RNAi/+;ppk^{1a}-Gal4,UAS-TBPH-Flag-HA/+*, SERCA Ri + Imp α3 Ri, *UAS-SERCA RNAi/UAS-Imp α3 RNAi;ppk^{1a}-Gal4,UAS-TBPH-Flag-HA/+*]. Outer and inner dashed lines indicate borders of cell bodies and nuclei, respectively (scale bar, 5 µm). (**F**) Quantification of Cyt/Nuc ratio of TBPH-Flag proteins in C4da neurons of Ctrl or expressing *SERCA Ri* or both *SERCA Ri* and *Imp α3 Ri* at 120 hr AEL. ****p<1.0×10$^{-4}$ by two-tailed t-test; error bars, ± SEM; n = 12 for Ctrl, n = 8 for *SERCA Ri*, n = 6 for *SERCA Ri + Imp α3 Ri*. (**G**) A schematic model for the regulatory mechanism of nucleocytoplasmic TBPH involving the cytosolic calcium-CalpA-Imp α3 pathway. Arrows indicate experimentally validated functional links, and dashed lines indicate possible alternative paths in addition to the validated cytosolic calcium-CalpA-Imp α3 pathway.

The online version of this article includes the following source data and figure supplement(s) for figure 5:

**Source data 1.** Numerical data plotted in *Figure 5B, D, F* and *Figure 5—figure supplement 1B*.
**Figure supplement 1.** Nuclear translocation of Imp α3 in C4da neurons overexpressing *CalpA*.

a mis-localization from the nucleus to the cytoplasm as observed in *Drosophila* retina (*Figure 6—figure supplement 3A–3C*) or from the cytoplasm to the nucleus as in C4da neurons.

## Increased cytosolic calcium restores defects in TBPH localization and larval locomotion in *C9orf72* ALS models

We showed above that altering the localization of TBPH can lead to an untimely functional output, leading to undesirable outcomes. In contrast to *Drosophila* larval C4da neurons, motor neurons in *Drosophila* larvae present predominantly nuclear TBPH. In motor neurons of ALS patients and model systems, depletion of TDP-43 in the nucleus (loss of function) and accumulation of TDP-43 in the cytoplasm (gain of toxicity)—both of which increase the Cyt/Nuc ratio of TDP-43—are crucial pathological features (*Neumann et al., 2006*; *Hergesheimer et al., 2019*). It has been reported that the increased Cyt/Nuc ratio of TDP-43 is observed in ~97% of ALS patients (*Ling et al., 2013*) and in ALS animal models (*Wils et al., 2010*; *Li et al., 2010*). These observations combined with our current data suggest a hypothesis that increasing cytosolic calcium may restore the dysregulated Cyt/Nuc ratio of TDP-43 in ALS. To test this hypothesis, we employed ALS fly models with motor neurons overexpressing arginine-rich dipeptide repeat proteins (DPRs: PR100 and GR100) derived from the GGGGCC repeat expansion in *C9orf72*, the most commonly mutated gene in ALS (*DeJesus-Hernandez et al., 2011*; *Renton et al., 2011*). We first confirmed that the Cyt/Nuc ratio of TBPH was increased in the *PR100-* and *GR100*-expressing larval motor neurons (*Figure 7A* left panels and 7B), compared to in the control (Ctrl), consistent with previous observation in adult fly brains (*Solomon et al., 2018*). We then increased the amount of cytosolic calcium in motor neurons of these ALS fly models by co-overexpressing *SERCA Ri* with *PR100* or *GR100*. Increased cytosolic calcium restored the Cyt/Nuc ratio of TBPH significantly (p<0.05) in the *PR100-* and *GR100*-expressing larval motor neurons (*Figure 7A* right panels and 7B).

Movement defects are one of the main pathological features in ALS. Thus, we next examined whether increasing cytosolic calcium can rescue larval locomotion defects derived from overexpression of *GR100* in motor neurons of the ALS fly model. We placed a larva in the center of the Petri dish containing 3% agar and then measured the total distance it travelled in 10 s. This experiment was carried out for *w1118* larvae overexpressing control vector (Ctrl), *SERCA Ri*, *GR100* and control vector (*GR100 O/E* + Ctrl), and *GR100* and *SERCA Ri* (*GR100 O/E* + *SERCA Ri*) (*Figure 7C*). After repeating the experiments using more than 23 different larvae for each genotype, we found that larvae overexpressing *SERCA Ri* showed no significant change in the distance travelled compared to the controls, whereas those overexpressing *GR100* showed a significant (p<0.05) decrease compared to the controls. The larvae co-overexpressing *GR100* and *SERCA Ri* travelled a longer distance than those expressing only *GR100* (*Figure 7D*), indicating mitigation of larval locomotion defect.

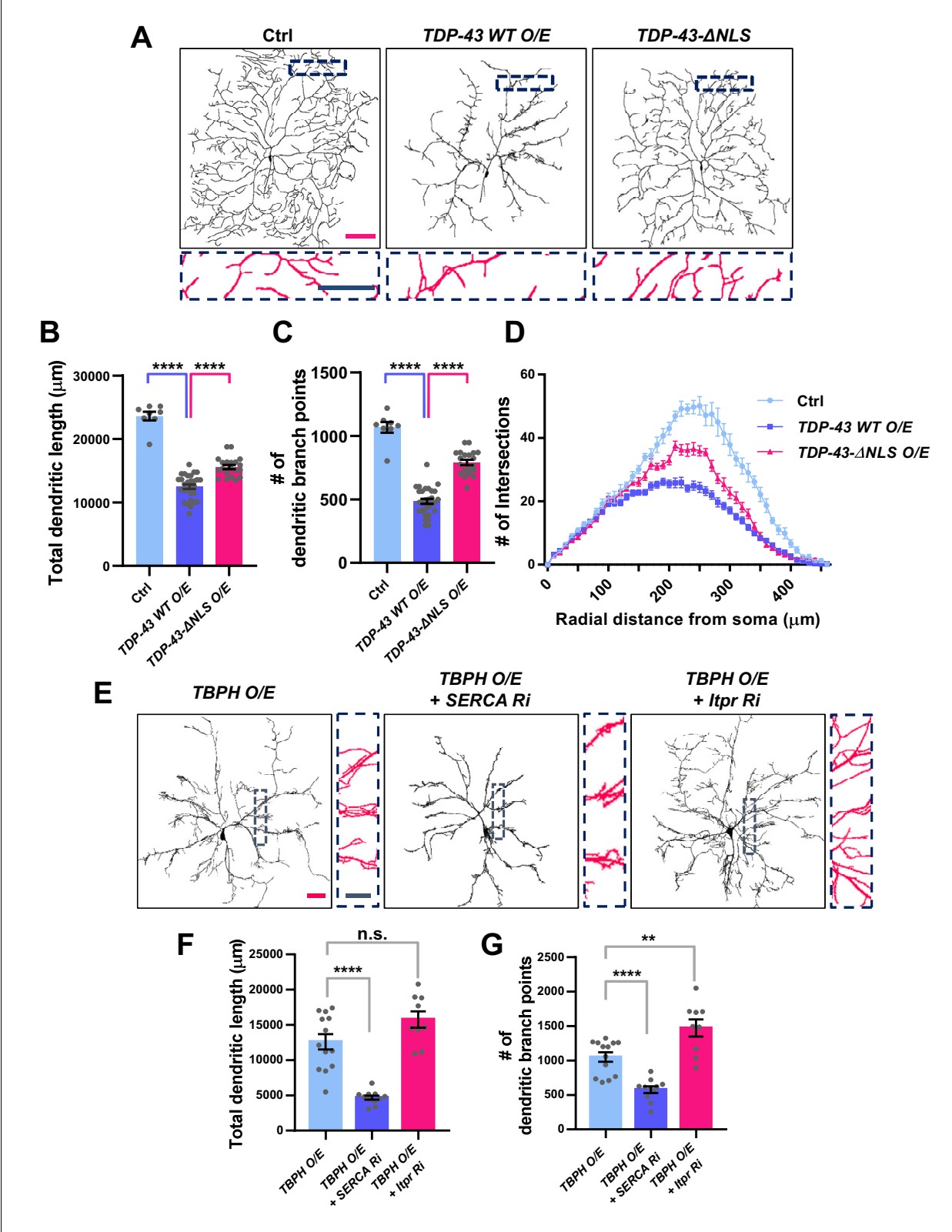

**Figure 6.** Significant alteration in the dendrite arborization of larval C4da neurons resulted from an untimely nuclear mis-localization of TBPH/TDP-43. (A) Skeletonized dendrite images of C4da neurons of larvae with denoted genotypes [Genotype: Ctrl, *+/+;ppk[1a]-Gal4,UAS-CD4-tdGFP/+, TDP-43 WT O/E, UAS-TDP-43/+;ppk[1a]-Gal4, UAS-CD4-tdGFP/+, TDP-43-ΔNLS O/E, UAS-TDP-43-ΔNLS/+;ppk[1a]-Gal4, UAS-CD4-tdGFP/+*]. Magnified images are presented at the bottom (Red scale bar, 100 μm; Blue scale bar, 50 μm). (B) Quantification of total dendritic length in neurons expressing the denoted

*Figure 6 continued on next page*

*Figure 6 continued*

transgenes. ****p<1.0×10⁻⁴ by two-tailed t-test; error bars, ± SEM; n = 8 for Ctrl, n = 20 for *TDP-43 WT O/E*, n = 12 for *TDP-43-ΔNLS O/E*. (C) Quantification of the number of dendritic branch points in neurons expressing the denoted transgenes. ****p<1.0×10⁻⁴ by two-tailed t-test; error bars, ± SEM; n = 8 for Ctrl, n = 20 for *TDP-43 WT O/E*, n = 12 for *TDP-43-ΔNLS O/E*. (D) Sholl analysis of neurons expressing the denoted transgenes. (E) Skeletonized dendrite images of C4da neurons of larvae with denoted genotypes [Genotype: *TBPH O/E, UAS-TBPH/+;ppk¹ᵃ-Gal4,UAS-CD4-tdGFP/+, TBPH O/E + SERCA Ri, UAS-TBPH/UAS-SERCA RNAi;ppk¹ᵃ-Gal4,UAS-CD4-tdGFP/+, TBPH O/E + Itpr Ri, UAS-TBPH/UAS-Itpr RNAi;ppk¹ᵃ-Gal4,UAS-CD4-tdGFP/+*]. Magnified images are presented at the right side (Red scale bar, 40 μm; Blue scale bar, 20 μm). (F) and (G) Quantifications of total dendritic length (F) and number of dendritic branch points (G) in neurons expressing the denoted transgenes. ****p<1.0×10⁻⁴, **p=0.0045, n.s., not significant, p=0.0655 by two-tailed t-test; error bars, ± SEM; n = 13 for *TBPH O/E*, n = 11 for *TBPH O/E + SERCA Ri*, n = 9 for *TBPH O/E + Itpr Ri*.

The online version of this article includes the following source data and figure supplement(s) for figure 6:

**Source data 1.** Numerical data plotted in *Figure 6B, C, F, G* and *Figure 6—figure supplement 1B, C, E, F* and *2D, E*.
**Figure supplement 1.** Dendritic defects induced by TBPH overexpression can be mitigated by reducing TBPH nuclear localization in C4da neurons.
**Figure supplement 2.** Restoration of dendrite arborization pattern in larval C4da neurons overexpressing *TBPH* by knockdown of a splicing factor *Hrb27C*.
**Figure supplement 3.** Cytoplasmic mis-localization of TDP-43 is associated with retinal degeneration.

Taken together, these data suggest that increasing cytosolic calcium can restore defects in TBPH localization and larval locomotion, at least in *C9orf72* ALS models.

## Discussion

In this study, we demonstrate a role of the calcium-CalpA-Imp α3 pathway in regulation of cytoplasmic accumulation of TDP-43. The involvement of this pathway further extends the previously proposed feedback cycle model (*Solomon et al., 2018*). In the previous model, amounts of cytoplasmic TDP-43 are increased through NCT defects caused by cytoplasmic aggregation of TDP-43, and increased levels of TDP-43 in the cytoplasm further accelerate TDP-43 aggregation. Our results provide an additional pathway that can increase the amounts of cytoplasmic TDP-43. How the calcium-CalpA-Imp α3 pathway interacts with the events in the feedback cycle model during progression of ALS is not certain. Interestingly, although modulation of cytosolic calcium levels led to reversible translocation of TDP-43 between the cytoplasm and the nucleus, TDP-43 aggregates were not observed in our study. Given this observation, the calcium-CalpA-Imp α3 pathway is likely to precede the sequestration of Importins by cytoplasmic TDP-43 aggregates and the subsequent NCT defects in the feedback cycle model. Therefore, our results provide an early regulatory mode to be therapeutically targeted for prevention of cytoplasmic accumulation of TDP-43 in ALS (*Figure 7—figure supplement 1*). In addition, we showed calcium-dependent dynamic changes of TDP-43 localization in neurons in response to the changes in cellular context during development. Together with a recent study showing shift in TDP-43 localization from nucleus to cytoplasm in skeletal muscle cells during regeneration (*Vogler et al., 2018*), our results suggest that translocation of TDP-43 is not necessarily pathological. This conclusion is supported by a recent study in which nuclear depletion and cytoplasmic accumulation of TDP-43 were observed in a subpopulation of neurons from various areas of healthy mouse brain (*Termsarasab et al., 2020*). These findings raise a new possibility that dynamic control of TDP-43 translocation can contribute to cytoplasmic accumulation of TDP-43 in ALS (*Figure 7—figure supplement 2*).

Among the candidate molecules selected from the genetic screen (*Figure 4* and *Figure 4—figure supplement 1*), we focused on CalpA and Imp α3 during the search of the mechanism. However, in addition to CalpA and Imp α3, the candidate molecules also included two other calcium-dependent regulator candidates (Cam and Pkc53E) and four other nuclear import components (Imp α1, Imp β1, Imp 7, and Tnpo-SR). Cam acts as a nuclear transporter of sex-determining region Y (SRY), independent of Importin (*Kaur et al., 2010*). However, Cam was excluded from the following experiments because of no Cam-binding site in TBPH, as well as the lethality induced by its overexpression. In contrast, Pkc53E regulates nucleocytoplasmic localization of diacylglycerol kinase ζ (*Topham et al., 1998*) and ribosomal protein S6 kinase (S6K) by inhibiting functions of their NLSs via phosphorylation (*Valovka et al., 2003*). Pkc53E showed a comparable effect to CalpA on the localization of TBPH (*Figure 4C*). These data suggest the existence of an additional, Pkc53E-dependent pathway to regulate TBPH localization. Imp αs are known to form heterodimers with Imp βs

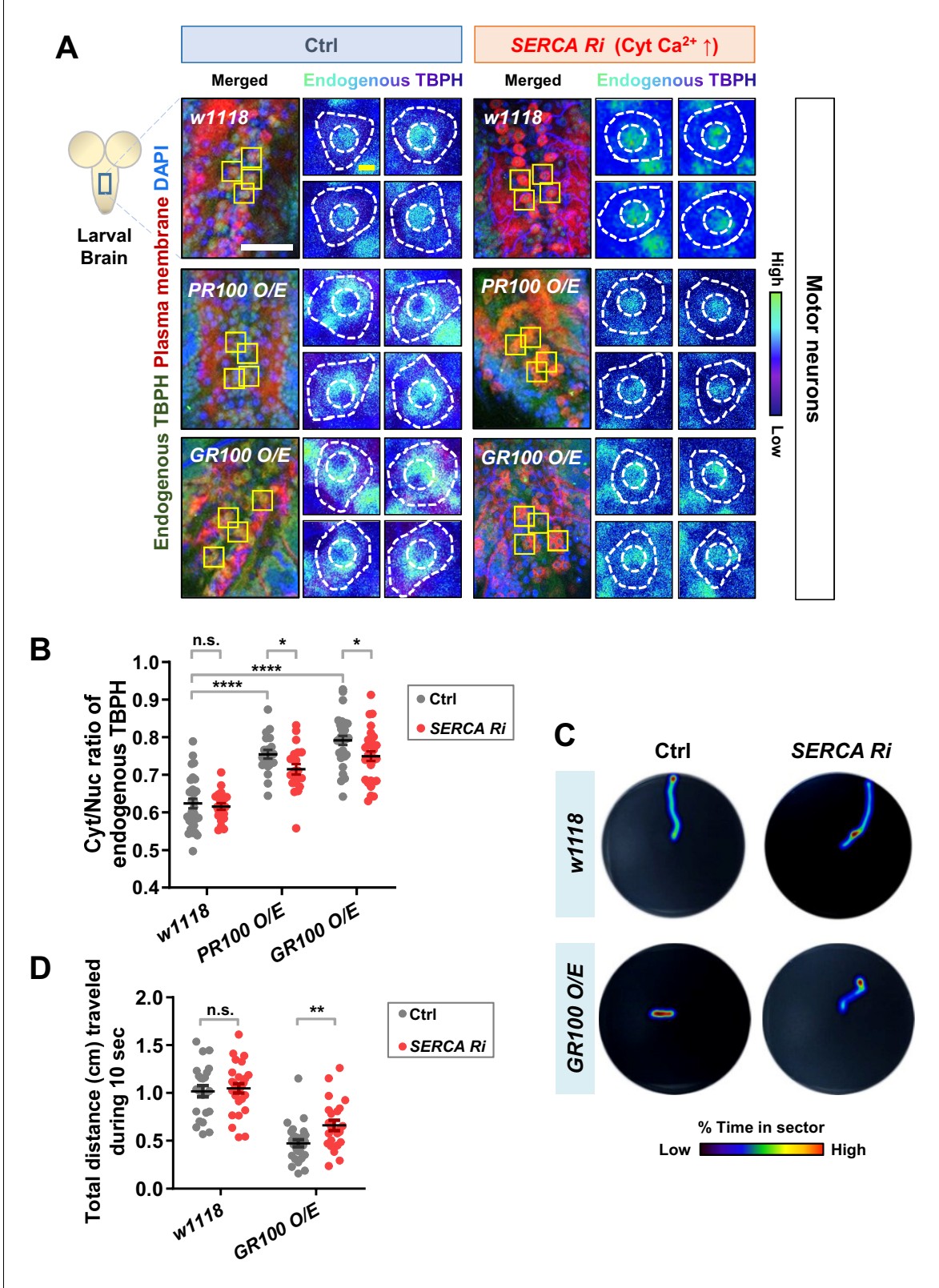

**Figure 7.** Restoration of aberrant TBPH localization and defective larval locomotion in *C9orf72* ALS models by increased cytosolic calcium. (A) Subcellular localization of endogenous TBPH proteins in larval motor neurons of *w1118* or overexpressing *PR100* or *GR100*, accompanied by concomitant expression of *40D$^{UAS}$* control vector (Ctrl) or *SERCA Ri* (*SERCA Ri*) at 120 hr AEL [Genotype: *w1118* in Ctrl, *40D$^{UAS}$/+;D42-Gal4,UAS-mCD8-RFP/+*, *PR100* O/E in Ctrl, *40D$^{UAS}$/UAS-poly-PR.PO-100;D42-Gal4,UAS-mCD8-RFP/+*, *GR100* O/E in Ctrl, *40D$^{UAS}$/UAS-poly-GR.PO-100;D42-Gal4,UAS-*

*Figure 7 continued*

*mCD8-RFP/+, w1118* in *SERCA Ri, UAS-SERCA RNAi/+;D42-Gal4,UAS-mCD8-RFP/+, PR100 O/E* in *SERCA Ri, UAS-SERCA RNAi/UAS-poly-PR.PO-100; D42-Gal4,UAS-mCD8-RFP/+, GR100 O/E* in *SERCA Ri, UAS-SERCA RNAi/UAS-poly-GR.PO-100;D42-Gal4,UAS-mCD8-RFP/+*]. Merged immunohistochemical images of endogenous TBPH proteins (green), plasma membranes (red), and DAPI (blue) are presented on the left side. Four neurons from the A3-A5 region of the VNC are selected as representative images (marked by yellow squares) and their enlarged images are shown on the right. Outer and inner dashed lines indicate borders of cell bodies and nuclei, respectively (yellow scale bar, 5 μm). (B) Quantification of Cyt/Nuc ratio of endogenous TBPH proteins in larval motor neurons expressing denoted transgenes described in A. N.S., not significant, p>0.05, ****$p<1.0\times10^{-4}$, *p<0.05 by two-tailed t-test; error bars, ± SEM; n = 30 for *w1118* in Ctrl, n = 20 for *PR100 O/E* in Ctrl, n = 30 for *GR100 O/E* in Ctrl; n = 20 for *w1118* in *SERCA Ri*, n = 20 for *PR100 O/E* in *SERCA Ri*, n = 30 for *GR100 O/E* in *SERCA Ri*. (C) Heat maps showing residence probability during traveling of larvae [Genotype: *w1118* in Ctrl, *40D^UAS^/+;D42-Gal4,UAS-mCD8-RFP/+, w1118* in *SERCA Ri, UAS-SERCA RNAi/+;D42-Gal4,UAS-mCD8-RFP/+, GR100 O/E* in Ctrl, *40D^UAS^/UAS-poly-GR.PO-100;D42-Gal4,UAS-mCD8-RFP/+, GR100 O/E* in *SERCA Ri, UAS-SERCA RNAi/UAS-poly-GR. PO-100;D42-Gal4,UAS-mCD8-RFP/+*] monitored in the 90 mm Petri dish until the larvae reached the edge of Petri dish or for up to 60 s. (D) Quantification of total distances traveled during 10 s for larvae expressing denoted genotypes in motor neurons described in C. n.s., not significant, p>0.05, **p=0.0048 by two-tailed t-test; error bars, ± SEM; n = 23 for *w1118* in Ctrl, n = 28 for *GR100 O/E* in Ctrl; n = 28 for *w1118* in *SERCA Ri*, n = 23 for *GR100 O/E* in *SERCA Ri*.

The online version of this article includes the following source data and figure supplement(s) for figure 7:

**Source data 1.** Numerical data plotted in *Figure 7B, D* and *Figure 7—figure supplement 4B*.
**Figure supplement 1.** A schematic illustration visualizing how our finding of the calcium-Calpain-A-Importin α3 pathway extends the current feedback cycle of TDP-43 aggregation and NCT defects during ALS pathogenesis.
**Figure supplement 2.** A schematic illustration showing the conceptual differences in understanding TDP-43 pathology between a previous model and our proposed model.
**Figure supplement 3.** Subcellular localization change of polyQ proteins and restoration of polyQ-induced retinal degeneration via knockdown of *CalpA*.
**Figure supplement 4.** *CalpA* overexpression increases the amount of TBPH protein without shifting its size.

(*Moroianu et al., 1995*, *Jäkel et al., 1999*) , suggesting that Imp α3 and Imp β1 may form a heterodimer to participate in our pathway. Indeed, disrupting this pathway via reducing *CalpA* level appears to interfere with nuclear translocation of another NLS-bearing protein, polyQ (MJDtr-78Q; *Figure 7—figure supplement 3A*), which is known to be transported via Imp α3 (*Sowa et al., 2018*). Notably, a reduction in *CalpA* mitigated polyQ-induced retinal degeneration (*Figure 7—figure supplement 3B*). These data suggest that calcium-mediated CalpA activity may affect nuclear localization of a broad range of proteins via Imp α3.

Calpains have been associated with neurotoxicity in a number of diseases including ALS (*Vosler et al., 2008*). There are 15 genes encoding calpains in mammals (*Evans and Turner, 2007*), among which Calpain-1 and −2 are most studied. In Neuro2a cells and in motor cortex of patients with frontotemporal lobar degeneration with clinical features of motor neuron disease (FTLD-MND), *Yamashita et al., 2012* showed that Calpain-1 and −2 are responsible for the calcium-mediated cleavage of TDP-43, leading to its toxic accumulation in the cytoplasm. Calpain-mediated cleavage of TDP-43 into CTF-25 and CTF-35 during traumatic brain injury in mice has also been reported (*Yang et al., 2014*). In *Drosophila*, CalpA is most similar to Calpain-9 (*Thurmond et al., 2019*). Therefore, some of the functions of Calpain-1 and −2, such as cleavage of TDP-43, may not be conserved in CalpA. To test whether CalpA used in this study can also cleave TBPH, we overexpressed *CalpA* in fly heads and measured the amount and size of TBPH via western blot and compared them to the control. We found that *CalpA* overexpression led to no apparent pattern of TBPH cleavage on the blot (*Figure 7—figure supplement 4A and B*), although unexpectedly, it led to an increase in the amount of TBPH. Instead, we showed that CalpA is necessary for nuclear import of TBPH (*Figure 4B and C*) through a yet undefined mechanism. These data suggest that CalpA, unlike Calpain-1 and −2, does not cleave TBPH and instead may be protective against TBPH-mediated toxicity.

Knockdown of *CalpA* blocked nuclear translocation of Imp α3 in pupal C4da neurons (*Figure 5A and B*), suggesting that CalpA functions as an upstream regulator of Imp α3. A specific mechanism by which CalpA regulates TBPH translocation through Imp α3 remains to be shown. Calpain, a human orthologue of CalpA, is a protease with a broad spectrum of substrates (*DuVerle et al., 2011*). However, Importin α3, a human orthologue of Imp α3, has not been reported as a substrate of Calpain, nor predicted using the Calpain cleavage detector (CCD) software (*Liu et al., 2011*). These data suggest the possibility that CalpA regulates nuclear translocation of Imp α3 indirectly via

its another target protein associated with nuclear import or export of Imp α3. Further functional study is needed to test the validity of this hypothesis.

Based on our data, Imp α3/β1 complex is the major transport regulator for nuclear import of TBPH (*Figure 4—figure supplement 2A and B*). Interestingly, knockdown of *Imp α1*, *Tnpo-SR*, and *Imp seven* also led to a significant shift in TBPH localization from the nucleus to the cytoplasm in pupae. TDP-43 has been shown to interact with both *KPNA4* and *KPNA6*, the orthologs for which are *Imp α3* and *Imp α1*, respectively (*Nishimura et al., 2010*). Interestingly, ectopic expression of *Imp α1* in *Drosophila* eyes rescued glassy-eye phenotype of *Imp α3* mutant (*Mason et al., 2003*), suggesting that Imp α1 and Imp α3 have somewhat redundant functions. Thus, it is not surprising that knockdown of *Imp α1* also disrupted nuclear transport of TBPH. Whether Imp α1 also receives regulation from CalpA remains to be shown. As for *Imp7* and *Tnpo-SR*, they both function at the upstream of mRNA splicing. *Imp7* binds directly to small nuclear ribonucleoproteins (snRNPs), which are indispensable constituents of a spliceosome, and localizes them into Cajal bodies (*Natalizio and Matera, 2013*). Now, spliceosome assembly is also mediated by serine/arginine-rich splicing factor 1 (SRSF1) (*Cho et al., 2011*), which is transported into the nucleus by Tnpo-SR (*Allemand et al., 2002*). Notably, 517 and 333 SRSF1-regulated alternative splicing events were identified in 3D MCF-10A acini and HeLa cells, respectively (*Anczuków et al., 2015*). We surmise that knockdown of *Imp7* and *Tnpo-SR* may cause widespread changes in alternative splicing, the result of which may impede TBPH import into the nucleus.

We reported in this study that TBPH localization shifts from the cytoplasm (at larval period) to the nucleus (at pupal period), but for what reason TBPH translocates into the nucleus during the pupal stage remains unknown. Interestingly, a previous study (*Vanden Broeck et al., 2013*) showed that loss of function of TBPH leads to increased *Map205* expression, resulting in cytoplasmic mis-localization of ecdysone receptor-A (EcR-A), an isoform of EcR expressed predominantly during the late stages of pupa. These results raise a possibility that in C4da neurons, TBPH translocates into the nucleus during pupal stage to regulate the expression of *Map205* and thus the nuclear localization of EcR-A, an important step for proper metamorphosis. Of note, several previous studies showed that the loss of liver X receptor β (LXRβ), a mammalian ortholog of EcR, results in motor neuron degeneration (*Andersson et al., 2005*; *Bigini et al., 2010*) and that LXRβ physically interacts with TDP-43 (*Vanden Broeck et al., 2013*). Another previous study showed that LXRβ links β-sitosterol to ALS-parkinsonism dementia complex (*Kim et al., 2008*). Furthermore, single-nucleotide polymorphisms (SNPs) for LXRβ were recently identified to be associated with the age of onset in ALS patients (*Mouzat et al., 2018*). These studies strongly suggest that disrupted LXRβ (EcR) signaling pathway may be associated with motor neuron diseases such as ALS. Taken together, it seems possible that neurons dynamically control the nucleocytoplasmic translocation of TDP-43 upon specific cellular demands such as regulation of LXRβ signaling associated with ALS pathogenesis.

Previous studies reported that increased calcium was observed in motor neurons of ALS mouse models (*von Lewinski and Keller, 2005*). Together with the fact that excessive intracellular calcium over the manageable capacity is able to trigger cell death, these observations support the idea that decreasing calcium can be an effective strategy to treat ALS. Consistent with this idea, riluzole, the FDA-approved medication for ALS, appears to delay the disease progression for 3 months by inhibiting calcium-mediated excitotoxicity. However, in a recent post-hoc study, the beneficial effect of riluzole was shown to be evident at late stage of ALS (stage 4), but not at earlier stages (stages 2 and 3) (*Fang et al., 2018*). Consistent with this post-hoc study, a previous study suggested that early intrinsic hyperexcitability may not contribute to motor neuron degeneration in ALS (*Leroy et al., 2014*). Moreover, a potentially protective function of calcium in motor neurons of ALS models at their early developmental stages has been reported (*Saxena et al., 2013*; *Armstrong and Drapeau, 2013*). Taken together, these findings suggest a possibility that intracellular calcium may have a concentration-dependent shift in its net effect on ALS pathogenesis from being protective to pathogenic as disease progresses. Our finding of the calcium-CalpA-Imp α3 pathway may provide a mechanistic insight to understand the role of calcium in ALS pathogenesis.

## Materials and methods

**Key resources table**

| Reagent type (species) or resource | Designation | Source or reference | Identifiers | Additional information |
|---|---|---|---|---|
| Genetic reagent (*D. melanogaster*) | *w1118* | Bloomington *Drosophila* Stock Center (BDSC) | RRID:BDSC_5905 | |
| Genetic reagent (*D. melanogaster*) | *D42-Gal4* | BDSC | RRID:BDSC_8816 | |
| Genetic reagent (*D. melanogaster*) | *TH-Gal4* | BDSC | RRID:BDSC_8848 | |
| Genetic reagent (*D. melanogaster*) | *repo-Gal4* | BDSC | RRID:BDSC_7415 | |
| Genetic reagent (*D. melanogaster*) | *UAS-mCD8-RFP* | BDSC | RRID:BDSC_27399 | |
| Genetic reagent (*D. melanogaster*) | *UAS-CD4-tdTom* | BDSC | RRID:BDSC_35841 | |
| Genetic reagent (*D. melanogaster*) | *Itpr$^{ka1091/+}$* | BDSC | RRID:BDSC_30739 | |
| Genetic reagent (*D. melanogaster*) | *Itpr$^{sv35/+}$* | BDSC | RRID:BDSC_30740 | |
| Genetic reagent (*D. melanogaster*) | *20XUAS-ChR2. T159C-HA* | BDSC | RRID:BDSC_52258 | |
| Genetic reagent (*D. melanogaster*) | *UAS-Chr2.s* | BDSC | RRID:BDSC_9681 | |
| Genetic reagent (*D. melanogaster*) | *RyR$^{16/+}$* | BDSC | RRID:BDSC_6812 | |
| Genetic reagent (*D. melanogaster*) | *UAS-NachBac* | BDSC | RRID:BDSC_9467 | |
| Genetic reagent (*D. melanogaster*) | *UAS-R-GECO1-IR1,UAS-R-GECO1.L-IR2* | BDSC | RRID:BDSC_52222 | |
| Genetic reagent (*D. melanogaster*) | *UAS-Cam RNAi* | BDSC | RRID:BDSC_34609 | |
| Genetic reagent (*D. melanogaster*) | *UAS-CanA-14F RNAi* | BDSC | RRID:BDSC_58249 | |
| Genetic reagent (*D. melanogaster*) | *UAS-CanB RNAi* | BDSC | RRID:BDSC_27307 | |
| Genetic reagent (*D. melanogaster*) | *UAS-CaMKI RNAi* | BDSC | RRID:BDSC_35362 | |
| Genetic reagent (*D. melanogaster*) | *UAS-Pka-C1 RNAi* | BDSC | RRID:BDSC_31599 | |

*Continued on next page*

*Continued*

| Reagent type (species) or resource | Designation | Source or reference | Identifiers | Additional information |
|---|---|---|---|---|
| Genetic reagent (*D. melanogaster*) | *UAS-Pkc53E RNAi* | BDSC | RRID:BDSC_55864 | |
| Genetic reagent (*D. melanogaster*) | *UAS-CalpA RNAi* | BDSC | RRID:BDSC_29455 | |
| Genetic reagent (*D. melanogaster*) | *UAS-CalpB RNAi* | BDSC | RRID:BDSC_25963 | |
| Genetic reagent (*D. melanogaster*) | *UAS-NFAT RNAi* | BDSC | RRID:BDSC_51422 | |
| Genetic reagent (*D. melanogaster*) | *UAS-nej RNAi* | BDSC | RRID:BDSC_37489 | |
| Genetic reagent (*D. melanogaster*) | *UAS-CrebA RNAi* | BDSC | RRID:BDSC_27648 | |
| Genetic reagent (*D. melanogaster*) | *UAS-CrebB RNAi* | BDSC | RRID:BDSC_63681 | |
| Genetic reagent (*D. melanogaster*) | *UAS-Imp α1 RNAi* | BDSC | RRID:BDSC_27523 | |
| Genetic reagent (*D. melanogaster*) | *UAS-Imp α2 RNAi* | BDSC | RRID:BDSC_27692 | |
| Genetic reagent (*D. melanogaster*) | *UAS-Imp α3 RNAi* | BDSC | RRID:BDSC_27535 | |
| Genetic reagent (*D. melanogaster*) | *UAS-Imp β1 RNAi* | BDSC | RRID:BDSC_31242 | |
| Genetic reagent (*D. melanogaster*) | *UAS-Imp 7 RNAi* | BDSC | RRID:BDSC_33626 | |
| Genetic reagent (*D. melanogaster*) | *UAS-Imp β11 RNAi* | BDSC | RRID:BDSC_55142 | |
| Genetic reagent (*D. melanogaster*) | *UAS-Tnpo RNAi* | BDSC | RRID:BDSC_50732 | |
| Genetic reagent (*D. melanogaster*) | *UAS-Tnpo-SR RNAi* | BDSC | RRID:BDSC_56974 | |
| Genetic reagent (*D. melanogaster*) | *UAS-Ran RNAi* | BDSC | RRID:BDSC_42482 | |
| Genetic reagent (*D. melanogaster*) | *UAS-Ntf-2 RNAi* | BDSC | RRID:BDSC_28633 | |
| Genetic reagent (*D. melanogaster*) | *UAS-Luciferase* | BDSC | RRID:BDSC_35788 | |

*Continued on next page*

*Continued*

| Reagent type (species) or resource | Designation | Source or reference | Identifiers | Additional information |
|---|---|---|---|---|
| Genetic reagent (*D. melanogaster*) | *UAS-Hrb87F RNAi* | BDSC | RRID:BDSC_52937 | |
| Genetic reagent (*D. melanogaster*) | *UAS-HNPNPC RNAi* | BDSC | RRID:BDSC_42506 | |
| Genetic reagent (*D. melanogaster*) | *UAS-glo RNAi* | BDSC | RRID:BDSC_33668 | |
| Genetic reagent (*D. melanogaster*) | *UAS-Syp RNAi* | BDSC | RRID:BDSC_56972 | |
| Genetic reagent (*D. melanogaster*) | *UAS-poly-PR. PO-100* | BDSC | RRID:BDSC_58698 | |
| Genetic reagent (*D. melanogaster*) | *UAS-poly-GR. PO-100* | BDSC | RRID:BDSC_58696 | |
| Genetic reagent (*D. melanogaster*) | *Gmr-Gal4* | BDSC | RRID:BDSC_1104 | |
| Genetic reagent (*D. melanogaster*) | *UAS-MJD-tr78Q* | BDSC | RRID:BDSC_8150 | |
| Genetic reagent (*D. melanogaster*) | *Df(2R)BSC26* | BDSC | RRID:BDSC_6866 | |
| Genetic reagent (*D. melanogaster*) | *UAS-SERCA RNAi* | Vienna *Drosophila* Resource Center (VDRC) | VDRC: 107446; RRID:FlyBase_FBst0479267 | |
| Genetic reagent (*D. melanogaster*) | *UAS-Itpr RNAi* | VDRC | VDRC: 106982; RRID:FlyBase_FBst0478805 | |
| Genetic reagent (*D. melanogaster*) | *UAS-RyR RNAi* | VDRC | VDRC: 109631; RRID:FlyBase_FBst0481295 | |
| Genetic reagent (*D. melanogaster*) | *UAS-CaMKII RNAi* | VDRC | VDRC: 100265; RRID:FlyBase_FBst0472139 | |
| Genetic reagent (*D. melanogaster*) | *UAS-Imp α3 RNAi* | VDRC | VDRC: 106249; RRID:FlyBase_FBst0478074 | |
| Genetic reagent (*D. melanogaster*) | *UAS-RanGAP RNAi* | VDRC | VDRC: 108264; RRID:FlyBase_FBst0480076 | |
| Genetic reagent (*D. melanogaster*) | *UAS-Rcc1 RNAi* | VDRC | VDRC: 110321; RRID:FlyBase_FBst0481896 | |
| Genetic reagent (*D. melanogaster*) | *UAS-Hrb98DE RNAi* | VDRC | VDRC: 29524; RRID:FlyBase_FBst0458009 | |
| Genetic reagent (*D. melanogaster*) | *UAS-Hrb27C RNAi* | VDRC | VDRC: 101555; RRID:FlyBase_FBst0473428 | |

*Continued on next page*

*Continued*

| Reagent type (species) or resource | Designation | Source or reference | Identifiers | Additional information |
|---|---|---|---|---|
| Genetic reagent (*D. melanogaster*) | *UAS-HNRNPU1 RNAi* | VDRC | VDRC:106984; RRID:FlyBase_FBst0478807 | |
| Genetic reagent (*D. melanogaster*) | *UAS-Sm RNAi* | VDRC | VDRC:108351; RRID:FlyBase_FBst0480162 | |
| Genetic reagent (*D. melanogaster*) | *UAS-CalpA RNAi* | VDRC | VDRC:101294; RRID:FlyBase_FBst0473167 | |
| Genetic reagent (*D. melanogaster*) | *40D$^{UAS}$* | VDRC | VDRC ID: 60101 | |
| Genetic reagent (*D. melanogaster*) | *UAS-TBPH-Flag-HA* | Bangalore Fly Resource Center | *Drosophila* Protein interaction Map (DPiM) | |
| Genetic reagent (*D. melanogaster*) | *ppk$^{1a}$-Gal4* | **Han et al., 2011**; Yuh Nung Jan (University of California, San Francisco (UCSF)) | | |
| Genetic reagent (*D. melanogaster*) | *UAS-tdTomato P2A GCaMP5G, attp1* | **Daniels et al., 2014**; Barry Ganetzky (University of Wisconsin-Madison) | | |
| Genetic reagent (*D. melanogaster*) | *UAS-3xMyc-RFP-TDP-43* | **Wang et al., 2011**; Brian D. McCabe, (Swiss Federal Institute of Technology (EPFL)) | | |
| Genetic reagent (*D. melanogaster*) | *UAS-TBPH* | **Wang et al., 2011**; Brian D. McCabe, (Swiss Federal Institute of Technology (EPFL)) | | |
| Genetic reagent (*D. melanogaster*) | *UAS-TDP-43 WT* | **Voigt et al., 2010**; Aaron Voigt (University Hospital, RWTH Aachen University) | | |
| Genetic reagent (*D. melanogaster*) | *UAS-TDP-43 G287S* | **Voigt et al., 2010**; Aaron Voigt (University Hospital, RWTH Aachen University) | | |
| Genetic reagent (*D. melanogaster*) | *UAS-Flag-TDP-43* | **Miguel et al., 2011**; Magalie Lecourtois (University of Rouen) | | |

*Continued on next page*

*Continued*

| Reagent type (species) or resource | Designation | Source or reference | Identifiers | Additional information |
|---|---|---|---|---|
| Genetic reagent (*D. melanogaster*) | *UAS-Flag-TDP-43-ΔNLS* | *Miguel et al., 2011*; Magalie Lecourtois (University of Rouen) | | |
| Genetic reagent (*D. melanogaster*) | *UAS-2xFlag-Imp α3 (vk00002)* | This paper | | SB Lab (DGIST) |
| Genetic reagent (*D. melanogaster*) | *UAS-V5-Imp β1 (vk00002)* | This paper | | SB Lab (DGIST) |
| Genetic reagent (*D. melanogaster*) | *UAS-CalpA-2xMyc (vk00002)* | This paper | | SB Lab (DGIST) |
| Genetic reagent (*D. melanogaster*) | *UAS-empty* | *Park et al., 2020* | | SB Lab (DGIST) |
| Antibody | Mouse monoclonal anti-Flag (DYKDDDDK) | Wako | Cat#: 012–22384; RRID:AB_10659717 | IHC (1:400) |
| Antibody | Rat monoclonal anti-HA | Roche | Cat#: 11867423001; RRID:AB_390918 | IHC (1:200) |
| Antibody | Rabbit anti-TBPH | LTK BioLaboratories, Taiwan; (*Lin et al., 2011*); C.-K. James Shen (Taipei Medical University) | | IHC (1:100) |
| Antibody | Rabbit polyclonal anti-TDP-43 | Proteintech | Cat#: 10782–2-AP, RRID:AB_615042 | IHC (1:400) |
| Antibody | Goat polyclonal anti-mouse Alexa Fluor 647 | Invitrogen | Cat#: A21236; RRID:AB_2535805 | IHC (1:400) |
| Antibody | Goat polyclonal anti-rat Alexa Fluor 647 | Jackson Immunoresearch Laboratories | Cat#: 112-605-003; RRID:AB_2338393 | IHC (1:200) |
| Antibody | Goat polyclonal anti-rabbit Alexa Fluor 647 | Invitrogen | Cat#: A21244; RRID:AB_2535812 | IHC (1:400) |
| Antibody | Goat polyclonal anti-HRP Alexa Fluor 488 | Jackson Immunoresearch Laboratories | Cat#: 123-545-021; RRID:AB_2338965 | IHC (1:400) |
| Antibody | Goat polyclonal anti-HRP Cy3 | Jackson Immunoresearch Laboratories | Cat#: 123-165-021; RRID:AB_2338959 | IHC (1:400) |
| Recombinant DNA reagent | Plasmid: *UAS-2xFlag-Imp α3* | This paper | | |
| Recombinant DNA reagent | Plasmid: *UAS-V5-Imp β1* | This paper | | |
| Recombinant DNA reagent | Plasmid: *UAS-CalpA-2xMyc* | This paper | | |
| Chemical compound, drug | All-*trans*-retinal (ATR) powder | Sigma-Aldrich | Cat#: R2500; CAS: 116-31-4 | 1 mM |

*Continued on next page*

Continued

| Reagent type (species) or resource | Designation | Source or reference | Identifiers | Additional information |
|---|---|---|---|---|
| Software, algorithm | Zen | Zeiss | RRID:SCR_013672 | |
| Software, algorithm | ImageJ | NIH | RRID:SCR_003070 | |
| Software, algorithm | ImageJ Ratio Plus (plug in) | NIH | | PMID:22051797 |
| Software, algorithm | GraphPad Prism | GraphPad Software | RRID:SCR_002798 | |
| Software, algorithm | EthoVision XT | Noldus Information Technology | RRID:SCR_000441 | |
| Software, algorithm | Adobe photoshop | Adobe | RRID:SCR_014199 | |
| Other | Flouro-Box | Neo Science | | FLB-001B |

### Drosophila melanogaster

Fly stocks used were as follows: *w1118, D42-Gal4, TH-Gal4, repo-Gal4, GMR-gal4, UAS-mCD8-RFP, UAS-CD4-tdTom, Itpr$^{ka1091/+}$, Itpr$^{sv35/+}$, 20XUAS-ChR2.T159C-HA, UAS-ChR2.S, RyR$^{16/+}$, UAS-Nach-Bac, UAS-R-GECO1-IR1,UAS-R-GECO1.L-IR2, MJDtr-78Q(s), elav-Gal4, UAS-Cam RNAi, UAS-CanA-14F RNAi, UAS-CanB RNAi, UAS-CaMKI RNAi, UAS-Pka-C1 RNAi, UAS-Pkc53E RNAi, UAS-CalpA RNAi (Ch.2), UAS-CalpA RNAi (Ch.3), UAS-CalpB RNAi, UAS-NFAT RNAi, UAS-nej RNAi, UAS-CrebB RNAi, UAS-Imp α1 RNAi, UAS-Imp α2 RNAi, UAS-Imp α3 RNAi, UAS-Imp β1 RNAi, UAS-Imp 7 RNAi, UAS-Imp β11 RNAi, UAS-Tnpo RNAi, UAS-Tnpo-SR RNAi, UAS-Ran RNAi, UAS-Ntf-2 RNAi, UAS-Hrb87F RNAi, UAS-HNPNPC RNAi, UAS-glo RNAi, UAS-Syp RNAi, Df(2R)BSC26, UAS-Luciferase, UAS-poly-PR.PO-100,* and *UAS-poly-GR.PO-100* were obtained from the Bloomington *Drosophila* Stock Center (BDSC). *UAS-SERCA RNAi, UAS-Itpr RNAi, UAS-RyR RNAi, UAS-CaMKII RNAi, UAS-Imp α3 RNAi, UAS-RanGAP RNAi, UAS-Rcc1 RNAi, UAS-Hrb98DE RNAi, UAS-Hrb27C RNAi, UAS-HNRNPU1 RNAi, UAS-Sm RNAi, UAS-CalpA RNAi* and *40D$^{UAS}$* were obtained from the Vienna *Drosophila* Resource Center (VDRC). *UAS-TBPH-Flag-HA* was obtained from the Bangalore Fly Resource Center. *ppk$^{1a}$-Gal4* (**Han et al., 2011**) was a gift from Yuh Nung Jan (UCSF). *UAS-tdTomato P2A GCaMP5G* (**Daniels et al., 2014**) was a gift from Barry Ganetzky (University of Wisconsin-Madison). *UAS-3xMyc-RFP-TDP-43* was a gift from Brian D. McCabe (EPFL). *UAS-Flag-TDP-43* and *UAS-Flag-TDP-43-ΔNLS* (**Miguel et al., 2011**) were gifts from Magalie Lecourtois (University of Rouen). *UAS-TDP-43 WT* and *UAS-TDP-43 G287S* were gifts from Aaron Voigt (University Hospital, RWTH Aachen University). *UAS-empty* (**Park et al., 2020**) was used as a control in *Drosophila* eye experiment. All Flies were raised at 27°C and 60% humidity.

### Generation of transgenic fly lines

*UAS-2xFlag-Imp α3* and *UAS-CalpA-2xMyc* transgenes were generated by using the LD13917 (Flybase ID: FBcl0163088) and LD22862 (Flybase ID: FBcl0178847) clones obtained from the *Drosophila* Genomics Resource Center (DGRC), respectively. *UAS-V5-Imp β1* transgene was synthesized by Genscript (USA). All these transgenes were subcloned into pACU2 vector, and the transgenic fly lines were generated by Bestgene Inc (USA).

### Immunohistochemistry

Larvae (120 hr AEL), pupae (18 hr APF), and adult flies (10d adult and 40d adult) were dissected in 1x Phosphate Buffered Saline (PBS) to obtain fillet or brain samples for immunohistochemical analyses. Obtained samples were fixed in 4% Paraformaldehyde for 20 min, washed in 0.3% PBST (Triton-X100 0.3% in PBS), and blocked in blocking buffer (5% Normal donkey serum or normal goat serum in 0.3% PBST) for 45 min at room temperature. Samples were then incubated with the following primary antibodies for overnight at 4°C: mouse anti-Flag (1F6, Wako; 1:400 dilution), rat anti-HA (3F10,

Roche; 1:200 dilution), rabbit anti-TBPH (1:100 dilution) (*Lin et al., 2011*), and goat anti-HRP Alexa Fluor 488 (Jackson Immunoresearch Laboratories; 1:400 dilution) antibodies. The next day, samples were washed for 10 min (repeated three times) in 0.3% PBST and incubated with the following secondary antibodies for 4 hr: goat anti-mouse Alexa Fluor 647 (Invitrogen; 1:400 dilution), goat anti-rat Alexa Fluor 647 (Jackson Immunoresearch Laboratories; 1:200 dilution), and goat anti-rabbit Alexa Fluor 647 (Invitrogen; 1:400 dilution) antibodies. Samples were then rinsed with 0.3% PBST for 10 min (repeated three times) and mounted with 70% glycerol in phosphate buffered saline (PBG) for imaging.

## Microscope image acquisition

All images were acquired using LSM 780, 800 (Zeiss) confocal microscope and Zen (Zeiss) software. All images of samples after immunohistochemistry experiments were taken at 200x and 400x magnifications using 20x and 40x objective lens, respectively. Images of the C4da sensory neurons were obtained from the abdominal segments A5-A6, where anterior is to the left and dorsal is up. Retinal images were obtained using Leica SP5. One-day-old adult fly eyes (left eyes only) were taken at 160x magnification immediately upon dissection.

## The ratiometric calcium imaging and analysis

For ratiometric calcium imaging of C4da neurons in larval (120 hr AEL) and pupal (18 hr APF) stages, genetically encoded calcium indicator (GCaMP) and red fluorescent proteins, tdTomato (tdTom), were co-expressed in C4da neurons using P2A system (*Daniels et al., 2014*). The pseudo-colored images reflecting relative GCaMP level to tdTom level were generated by ImageJ Ratio plus. The mean pixel intensity of GCaMP/tdTom ratio was then measured using ImageJ to determine relative calcium levels, as previously described (*Kardash et al., 2011*).

## Optogenetic stimulation

Larvae expressing channelrhodopsin in C4da neurons were used for optogenetic experiments (behavioral assay and calcium imaging). Larvae were raised under constant darkness at 27°C and 60% humidity on standard media containing 1 mM ATR (Sigma-Aldrich) and collected at 5 days AEL. ATR inhibits the closed state of cyclic nucleotide–gated channels, including channelrhodopsin: ATR is needed to keep the channels open upon stimulation with blue light. Optogenetic stimulation (470 nm) was achieved by Flouro-Box (FLB-001B). Illumination duration and frequency (denoted conditions in *Figure 2G* and *Figure 2—figure supplement 1H*) were controlled manually. Both control and experimental groups were placed in the box for illumination, but control groups were prevented from light exposure by covering the vials with aluminum foil. Optogenetic stimulation was performed in a room temperature of 25°C. The validity of optogenetic stimulation and concomitant calcium uptake of C4da neurons was confirmed by both live imaging of calcium indicator RGECO1 and monitoring nociceptive rolling behavior (*Kaneko et al., 2017*).

## Fluorescence recovery after photobleaching (FRAP) experiment and analysis

FRAP experiment was performed by using Zeiss confocal microscope (LSM800). The control and experimental larvae were fed food without and with ATR, respectively. The optogenetic stimulation protocol from *Figure 2—figure supplement 1H* was applied to the larvae prior to FRAP. Nuclear area was selected as a region of interest (ROI), and cytoplasmic and extracellular areas of the cell were selected as reference and background, respectively, for normalization. Three pre-bleach images were obtained, and then photobleaching of the ROI was performed with 100% power of DPSS laser (561 nm laser) for three iterations. Images for fluorescence recovery were taken every 2 s during 5 min. For quantitative analysis, first, the mean intensity values of background were subtracted from those of ROI in obtained images. Then, these subtracted values were normalized to the fluorescence intensity of the reference. The normalized values were plotted for comparison, and best-fit curve was applied to the graph by using non-linear regression.

### Quantitative analysis of dendrites

All images of dendrites were subjected to skeletonization using ImageJ for subsequent analyses of dendritic length and number of branch points. Sholl analysis protocol was adapted from a previous study (*Yadav et al., 2019*).

### Larval motility assay and analysis

The wandering larvae (120 hr AEL) expressing denoted transgenes in motor neurons using the *D42-Gal4* driver were used for motility assays. Prior to the assay, individual larva was gently washed in 1x PBS, then briefly placed in a 90 mm Petri dish containing 25 ml of 3% agar. The larva was then placed in another identical Petri dish inside a dark box equipped with indirect lighting. For each genotype, the time for larvae (n ≥ 23 per genotype) to reach the edge of the Petri dish or for up to 60 s was recorded. Recording only started after larva's first sign of forward movement. These steps were applied to all genotypes tested.

To analyze the motility assay, EthoVision XT (13 version; Noldus Information Technology) video tracking system was used (*Noldus et al., 2001*). Residence probability until reaching the edge of the dish (up to 60 s) was shown as heat map (the color represented the amount of resident time; Red: large, Blue: small). Total distance travelled by larvae during 10 s was analyzed.

### Western blot

Fly head samples were prepared in a lysis buffer solution (50 mM Tris-buffered saline (Tris-HCL) pH 7.5, 150 mM NaCl, 1% Triton X100) with protease inhibitor cocktail (Thermo Scientific, #87786; 1:100). Samples were centrifuged for 15 min at 13,300 rpm and the supernatants were collected into new tubes. Subsequently, their protein amount was measured using Bradford protein assay. Quantified proteins were mixed with a solution containing 9:1 ratio of Laemmli buffer (Bio-Rad, #161–0747) to 2-mercaptoethanol (BIOSESANG, #60-24-2) and were boiled at 95℃ for 5 min. Samples were then loaded onto Mini-PROTEAN TGX Stain-Free, 4–15% gel (#BR456-8083, Bio-Rad). Protein transfer to the membrane (PVDF) was followed by an incubation in 5% skim milk diluted in 1% TBST (blocking buffer) for 1 hr at RT. Then they were incubated with primary antibodies overnight at 4℃. The follow primary antibodies were used: Rabbit anti-TBPH (from Dr. C.-K. James Shen) (1:5000), Rat anti-elav (DSHB 7E8A10) (1:200,000). After washing, the membranes were then incubated for 1 hr at RT with the corresponding secondary antibodies: Goat anti-rat IgG-HRP (Santa Cruz, sc-2006) (1:100,000) and Goat anti-rabbit IgG HRP (Santa Cruz, sc-3837) (1:5000). Finally, after washing five times in 1% TBST at RT, the membranes were incubated with ECL solution prior to detection using ChemiDocTM XRS+.

### Quantification and statistical analysis

To calculate the cytoplasm-to-nucleus (Cyt/Nuc) ratios of immunostained proteins (TBPH-Flag-HA, endogenous TBPH, 3xMyc-RFP-TDP-43, and 2xFlag-Imp α3), the mean pixel intensities of them in the nucleus and cytoplasm were measured using ImageJ (NIH) and Adobe photoshop (Adobe).

Statistical analysis was performed using GraphPad Prism (GraphPad Software), with Student's t-test and one-way ANOVA followed by Tukey's post hoc analysis. In all figures, N.S., *, **, ***, and **** represent $p>0.05$, $p<0.05$, $p<0.01$, $p<1.0\times10^{-3}$, and $p<1.0\times10^{-4}$ respectively. Error bars are standard errors of the mean (SEM).

## Acknowledgements

We thank the Bloomington *Drosophila* Stock Center, the Vienna *Drosophila* Resource Center and Bangalore Fly Resource Center for fly stocks. We thank Yuh Nung Jan (UCSF), Barry Ganetzky (University of Wisconsin-Madison), Aaron Voigt (University Hospital, RWTH Aachen University), Brian D McCabe (EPFL) and Magalie Lecourtois (University of Rouen) for providing precious resources. This work was supported by Basic Science Research Program through the National Research Foundation of Korea, funded by the Ministry of Science and Information and Communications Technology (ICT) (2018R1A2B6001607, 2019R1A4A1024278); the Development of Platform Technology for Innovative Medical Measurements Program from the Korea Research Institute of Standards and Science Grant (KRISS-2019-GP2019-0018); KBRI basic research program through Korea Brain Research Institute

funded by Ministry of Science and ICT (20-BR-04–02) (to SB Lee); and Institute for Basic Science Grant (IBS-R013-A1) funded by the Ministry of Science and ICT (to D Hwang).

## Additional information

### Funding

| Funder | Grant reference number | Author |
| --- | --- | --- |
| Ministry of science and ICT | 2018R1A2B6001607 | Sung Bae Lee |
| Ministry of science and ICT | 2019R1A4A1024278 | Sung Bae Lee |
| Korea Research Institute of Standards and Science | KRISS-2019-GP2019-0018 | Sung Bae Lee |
| Ministry of Science and ICT | 20-BR-04-02 | Sung Bae Lee |
| Ministry of Science and ICT | IBS-R013-A1 | Daehee Hwang |

The funders had no role in study design, data collection and interpretation, or the decision to submit the work for publication.

### Author contributions

Jeong Hyang Park, Chang Geon Chung, Conceptualization, Resources, Formal analysis, Investigation, Visualization, Writing - original draft, Writing - review and editing; Sung Soon Park, Davin Lee, Formal analysis, Investigation, Visualization, Writing - original draft; Kyung Min Kim, Formal analysis, Investigation, Visualization; Yeonjin Jeong, Formal analysis, Visualization; Eun Seon Kim, Yu-Mi Jeon, Hyung-Jun Kim, Investigation; Jae Ho Cho, Formal analysis; C-K James Shen, Resources; Daehee Hwang, Conceptualization, Supervision, Funding acquisition, Writing - original draft, Writing - review and editing; Sung Bae Lee, Conceptualization, Resources, Supervision, Funding acquisition, Writing - original draft, Writing - review and editing

### Author ORCIDs

Jeong Hyang Park (ID) https://orcid.org/0000-0002-7392-8366
Chang Geon Chung (ID) https://orcid.org/0000-0001-8155-4926
Sung Bae Lee (ID) https://orcid.org/0000-0002-8980-6769

### Decision letter and Author response

Decision letter https://doi.org/10.7554/eLife.60132.sa1
Author response https://doi.org/10.7554/eLife.60132.sa2

## Additional files

### Supplementary files

• Transparent reporting form

### Data availability

All data generated or analysed during this study are included in the manuscript and supporting files.

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
