## [Decision Letter]

**Acceptance summary:**

Cytoplasmic accumulation of TDP-43 is the most prominent cellular characteristic for amyotrophic lateral sclerosis (ALS). However, how TDP-43 shuttles between the nucleus and cytoplasm and how TDP-43 remains in the cytoplasm during pathological conditions have not been thoroughly investigated. In this study, Lee and colleagues discovered that TBPH, a TDP-43 orthologue in *Drosophila*, naturally translocates between the nucleus and cytoplasm during development and identified the calcium-Calpain-A-Importin α3 as a regulatory pathway for the TBPH movement. These findings provide new insights into the cellular behavior of TDP-43 and raise a possibility that dynamic control of TDP-43 translocation can contribute to excessive TDP-43 accumulation in ALS.

**Decision letter after peer review:**

Thank you for submitting your article "Cytosolic calcium regulates cytoplasmic accumulation of TDP-43 through Calpain-A and Importin α3" for consideration by *eLife*. Your article has been reviewed by three peer reviewers, one of whom is a member of our Board of Reviewing Editors, and the evaluation has been overseen by K VijayRaghavan as the Senior Editor. The following individual involved in review of your submission has agreed to reveal their identity: Randal S Tibbetts (Reviewer #3).

The reviewers have discussed the reviews with one another and the Reviewing Editor has drafted this decision to help you prepare a revised submission.

Summary:

In this manuscript, Lee and colleagues investigated molecular mechanisms underlying the nucleocytoplasmic transport of TDP-43 in *Drosophila* sensory neurons and identified cytosolic calcium-CalpainA-Importin a3 as a key regulatory axis for the transient localization of TDP-43. The authors first confirmed that the localization of TDP-43 changes over larval-pupal-adult development whose pattern can be significantly altered by changing the level of cytosolic calcium. Moreover, the authors took on to screen genetic modifiers of the TDP-43 shuttling and found two key genes, CalpainA and Imp-α3, in the process. Alterations in the TDP-43 localization in C4 neurons largely inhibit the dendritic arborization, which is partially rescued by knocking down Impa3. Further, the authors verified the similar phenomenon in the C9orf72 ALS model, extending their findings to the clinical perspective and suggest the Ca^2+^-CalpainA-Impa3 axis as a potential therapeutic target for ALS.

All the reviewers found the work interesting and well-executed, and most of the conclusions are convincing. While the developmental studies are clearly presented, the part where the authors describe a link to human pathomechanism leaves room to improve presented data and overall novelty. Several revisions were requested, which may require additional experiments.

Essential revisions:

1) Improve the quality of Figure 7. Images in Figure 7A can be better presented. Merged VNC images do not provide precise information as to what the authors want to show. Selected motor neurons in yellow insets are in variable locations and do not seem to indicate identical neurons. Although quantified, one or more motor neurons in the same position should be presented.

2) Show developmental or stage-specific expressions of TBPH, and discuss their biological meanings. What is TBPH localization at intermediate stages of development prior to pupation? Amongst many developmental stages, the authors focused on 120h AEL and 18h APF without providing backgrounds or rationales. Why are the stages critical for the experiment?

3) Provide additional descriptions of the optogenetic experiments; especially two different exposure times in Figure 2/Figure 2—figure supplement 1. An additional description of the optogenetic approach is needed in the text.

4) Confirm whether C4da specific knockdown of calcium modulators (e.g. Iptr) leads to the same TBPH translocation and whether it can rescue dendritic arborization phenotypes caused by TBPH. It would seem important to show that Ca^2+^-handling mutants (Iptr mutant or RNAi background) also impact TBPH toxicity using dendritic arborization, larval mobility, or other assays.

5) The authors claim in Figure 6 that nuclear TBPH is toxic while they conclude that increasing nuclear TBPH is protective of C9 DPR in Figure 7. The authors need to explain these contradicting explanations and provide additional data to support the claims. All experiments were carried out using fly TBPH. While this approach holds advantages, it would be important to compare wild-type and ALS-mutant human TDP-43 proteins for their nucleocytosolic shuttling dynamics in a subset of the developmental and Ca2-mobilization studies.

---

## [Author Response]

Essential revisions:1) Improve the quality of Figure 7. Images in Figure 7A can be better presented. Merged VNC images do not provide precise information as to what the authors want to show. Selected motor neurons in yellow insets are in variable locations and do not seem to indicate identical neurons. Although quantified, one or more motor neurons in the same position should be presented.

Based on the reviewers’ suggestion, we modified how we present the images in Figure 7A. To provide a more precise information, we enlarged the merged VNC images. This allowed us to mark more precisely some of the neurons that we quantified. For the ease of comparison, we have selected neurons from the same location (A3-A5) in the VNC and presented them in Figure 7A as representative images.

2) Show developmental or stage-specific expressions of TBPH, and discuss their biological meanings. What is TBPH localization at intermediate stages of development prior to pupation? Amongst many developmental stages, the authors focused on 120h AEL and 18h APF without providing backgrounds or rationales. Why are the stages critical for the experiment?

In Figure 1—figure supplement 1, we showed that the localization of endogenous TBPH shifts from the cytoplasm in larval C4da neurons to the nucleus in pupal C4da neurons. By comparing the level of immunostained endogenous TBPH between larval and pupal C4da neurons, we can speculate that the expression of TBPH does not change much between the two stages. However, we need to be extra cautious when comparing TBPH expression based on the immunostaining data, because the efficiency of antibody staining likely differs between the two stages. Furthermore, the difference in C4da neuron size between the two stages may create complications when analyzing the expression level via immunostaining. Endogenous TBPH staining also lead to the detection of TBPH in the tissues around C4da neurons including epithelia, muscles, and glia, all of which obstruct our ability to detect TBPH in C4da neuronal dendrites. Doing a western blot will require isolation of C4da neurons due to the cell-type specific effects observed in our study. Because of the small number of C4da neurons per larva, it will take a herculean effort to collect enough samples for a western blot. Thus, the technical difficulties greatly limit our ability to compare developmental or stage-specific expressions of TBPH with precision.

To test whether developmental processes might affect TBPH localization, we thought that it was important to examine TBPH localization at three major stages of a fly—larva, pupa, and adult (sub-divided into early and late adult)—because the cellular condition changes dramatically from one stage to the next, which may affect TBPH localization. This hypothesis was confirmed by the change in the intracellular calcium level between larval and pupal C4da neurons, which led to the altered localization of TBPH in pupa. As recommended by the reviewer, we also examined the localization pattern of TBPH at the prepupa stage (0h APF), the intermediate stage between larva and pupa (see revised Figure 1C and D). Our results showed an intermediate TBPH localization pattern at the intermediate stage, suggesting that TBPH undergoes a localization transition from the cytoplasm into the nucleus in prepupa. Although we initially examined three major developmental stages of a fly in Figure 1, we decided to focus more specifically on larval (120h AEL) and pupal (18h APF) stages for a couple of reasons. First, larval C4da neurons showed the most dramatic cytoplasmic localization of TBPH among all the stages examined, thus rendering this stage an ideal timepoint at which to examine the mechanisms underlying the cytoplasmic localization of TBPH. 18h APF—but not 40d adult—was chosen as the preferred stage to model nuclear localization of TBPH because it is temporally closer to the larval stage than 40d adult stage is. Choosing these two consecutive stages for further studies in our manuscript helped us to save time as we did not need to wait 40d when probing mechanisms underlying the stage-dependent nucleocytoplasmic translocation of TBPH.

What biological meaning might underlie the observed stage-specific alteration of the TBPH expression pattern? This stage-specific alteration can be speculated as one way by which the neurons regulate TBPH function. For instance, we reported that TBPH localizes in the cytoplasm of C4da neurons at the larval stage, unlike in other cell types (motor and DA neurons and glial cells). Nuclear TBPH in larval C4da neurons induced a significant dendrite toxicity compared to cytoplasmic TBPH. These data suggest that limiting nuclear function of TBPH in C4da neurons at the larval stage is needed to maintain neuronal health. Why does TBPH then translocate from the cytoplasm to the nucleus at the pupal stage? Previously, loss of TBPH function was shown to increase *Map205* expression, resulting in cytoplasmic mis-localization of ecdysone receptor-A (EcR-A), which is expressed predominantly during the late pupa stage (Vanden Broeck et al., 2013). According to this result, TBPH may translocate into the nucleus during the pupal stage to regulate *Map205* expression and nuclear localization of EcR-A, which is important for proper metamorphosis. Of note, loss of liver X receptor β (LXRβ), a mammalian ortholog of EcR, results in motor neuron degeneration (Andersson et al., 2005; Bigini et al., 2010) and LXRβ physically interacts with TDP-43 (Vanden Broeck et al., 2013). Taken together, neurons may control dynamically the nucleocytoplasmic translocation of TDP-43 upon specific cellular demands such as regulation of EcR-A localization during metamorphosis.

The newly added results and the discussion on the potential biological meaning of the differences in TBPH localization between stages have been incorporated into our revised manuscript.

Results:

“The endogenous TBPH expression level in C4da neurons between the two stages seemed comparable based on the immunostaining data”.

“TBPH was localized to both the cytoplasm and the nucleus in an intermediate stage (0h APF) between larva and pupa (Figure 1C), suggesting that TBPH may begin its nuclear entry at this stage. […] Quantitative comparisons revealed that Cyt/Nuc ratios between pairs of consecutive developmental stages (AEL-0h APF, 0h-18h APF, 18h APF-early adult, and early-late adult) were significantly different (Figure 1D)”.

Discussion

“We reported in this study that TBPH localization shifts from the cytoplasm (at larval period) to the nucleus (at pupal period), but for what reason TBPH translocates into the nucleus during the pupal stage remains unknown. […] Taken together, it seems possible that neurons dynamically control the nucleocytoplasmic translocation of TDP-43 upon specific cellular demands such as regulation of LXRβ signaling associated with ALS pathogenesis”.

3) Provide additional descriptions of the optogenetic experiments; especially two different exposure times in Figure 2/Figure 2—figure supplement 1. An additional description of the optogenetic approach is needed in the text.

In the original manuscript, we showed two separate optogenetic data (one in Figure 2G-I and another in Figure 2—figure supplement 1H-J) using two different experimental protocols with distinct exposure times. As suggested by the reviewer, this difference in protocols needs to be better described and justified. However, a better solution might be to apply the same protocol for both experiments. We thus performed the experiments for Figure 2—figure supplement 1 with the same protocol used for Figure 2G-I and replaced the previous results with the new ones in the revised manuscript. Furthermore, as requested, we have now incorporated additional descriptions of the optogenetic experiments. We added several details such as channelrhodopsin expression and the feeding of all trans-retinal (ATR) in larvae prior to the optogenetic experiment. We hope that the description of the optogenetic experiments are sufficiently clear for the reviewers and the general readership. The excerpt describing the optogenetic experiments from Figure 2/Figure 2—figure supplement 1 in the revised manuscript is provided below:

Results

“To reduce the possibility of activating compensatory mechanism from chronically increasing cytosolic calcium, we decided to increase intracellular calcium level in larval C4da neurons using an optogenetics technique. […] When we decreased the total amount of larval exposure to blue light to just 5 min, the Cyt/Nuc ratio of RFP-TDP-43 did not change at any of the time points (1 min, 30 min, 60 min, 90 min, and 120 min) imaged after the exposure (data not shown), suggesting that five minutes of increased cytosolic calcium is not enough to alter RFP-TDP-43 localization”.

4) Confirm whether C4da specific knockdown of calcium modulators (e.g. Iptr) leads to the same TBPH translocation and whether it can rescue dendritic arborization phenotypes caused by TBPH. It would seem important to show that Ca^2+^-handling mutants (Iptr mutant or RNAi background) also impact TBPH toxicity using dendritic arborization, larval mobility, or other assays.

We thank the reviewers for an invaluable experimental suggestion. A knockdown of *Itpr* or another *Itpr* mutant (*Itpr^sv35^*^/+^) showed a significantly higher amount of TBPH in the cytoplasm than in the nucleus of pupal C4da neurons, resulting in an increased Cyt/Nuc ratio compared to the controls (Figure 2—figure supplement 1A and B). These data are consistent with the *Itpr^ka1091/+^*pupae data (Figure 2C and D) from our original manuscript.

Next, we tested whether Ca^2+^-handling mutants or knockdowns also impact TBPH-induced alteration in dendritic arborization pattern. We have shown in Figure 6 of the original manuscript that nuclear TBPH is toxic and reduces total dendritic length and the number of dendritic branch points in C4da neurons. In the revised manuscript, we show that *SERCA* knockdown, which promotes nuclear translocation of overexpressed TBPH, exacerbated the TBPH-induced toxicity by reducing the total dendritic length and the number of dendritic branch points in larval C4da neurons (Figure 6E-G). In addition, *Itpr* knockdown (Figure 6E-G) or mutants (Figure 6—figure supplement 1A-C), which inhibit nuclear translocation of overexpressed TBPH, mitigated TBPH-induced dendrite toxicity by increasing the total dendritic length and the number of dendritic branch points. Together, these data suggest that intracellular calcium not only regulates the nucleocytoplasmic localization of TBPH, but also modulates TBPH-mediated dendritic defects. These additional results are now incorporated into the revised manuscript.

Results

“Similar results were found using another *Itpr* mutant (*Itpr^sv35^*^/+^) and *Itpr* RNAi (Figure 2—figure supplement 1A and B)”.

“Consistently, manipulating cytosolic calcium to alter the localization of overexpressed TBPH also led to similar dendritic phenotypes. *SERCA* knockdown, which promotes nuclear translocation of overexpressed TBPH, decreased the total dendritic length (Figure 6E and F) and the number of dendritic branch points (Figures 6E and G) in larval C4da neurons. *Itpr* knockdown (Figure 6E and G) or mutants (Figure 6—figure supplement 1A-1C), which inhibit nuclear translocation of overexpressed TBPH, increased the total dendritic length and the number of dendritic branch points.”

5) The authors claim in Figure 6 that nuclear TBPH is toxic while they conclude that increasing nuclear TBPH is protective of C9 DPR in Figure 7. The authors need to explain these contradicting explanations and provide additional data to support the claims. All experiments were carried out using fly TBPH. While this approach holds advantages, it would be important to compare wild-type and ALS-mutant human TDP-43 proteins for their nucleocytosolic shuttling dynamics in a subset of the developmental and Ca2-mobilization studies.

We claimed that nuclear TBPH is toxic in C4da neurons (Figure 6) because the original location of TBPH is mostly cytoplasmic in C4da neurons. However, we claimed that cytoplasmic TBPH is toxic in motor neurons because the original location of TBPH is nuclear in those neurons. The hypothesis that underlies these claims is that if the localization of TBPH is altered from its original cell-type specific localization, then that would induce toxicity. Indeed, C4da neurons overexpressing TBPH in the cytoplasm showed considerably less dendritic defects than those overexpressing it in the nucleus (Figure 6). In motor neurons, we showed that *SERCA* knockdown reduced cytoplasmic mis-localization of TBPH and mitigated the locomotion defect of GR100-expressing larvae, indicating a correlation between the cytoplasmic TBPH in motor neurons and the locomotion defect (Figure 7). In the revised manuscript, we further tested our hypothesis in another cell type, the retina. First, we examined the localization of endogenous TBPH in the neurons of eye imaginal disc in 3^rd^ instar larvae and showed that it is mostly nuclear. TDP-43 WT expressed in these cells was shown to be mostly confined to the nucleus, whereas TDP-43-∆NLS was observed mostly in the cytoplasm. At 1-day post eclosion, we noticed no degenerative phenotypes in flies expressing TDP-43 WT. On the other hand, TDP-43-∆NLS expression induced a significant retinal degeneration. These results support our hypothesis that altered localization of TBPH/TDP-43 from its original cell-type specific localization is toxic. These additional results and the corresponding text are now incorporated into the revised manuscript.

In our original manuscript, most of the experiments on nucleocytoplasmic shuttling have been carried out using *Drosophila* TDP-43 (TBPH). As suggested by the reviewers, we compared WT and ALS-mutant human TDP-43 for their nucleocytoplasmic shuttling dynamics in a subset of the developmental and calcium-mobilization studies. First, we expressed human TDP-43 WT and ALS-mutant TDP-43 G287S in C4da neurons and then examined their localization patterns along the development from larva to pupa. As expected, the localization of TDP-43 WT significantly changed from the cytoplasm at the larval stage (120h after egg laying; 120h AEL) to the nucleus at the pupal stage (18h after puparium formation; 18h APF). The localization of TDP-43 G287S also shifted from the cytoplasm to the nucleus, albeit less so compared to TDP-43 WT. Notably, the net change in the Cyt/Nuc ratio from larvae to pupae decreased more for TDP-43 than for TDP-43 G287S. When we decreased cytosolic calcium via *ryanodine receptor* (*RyR*) knockdown in pupae, the Cyt/Nuc ratio of TDP-43 WT increased significantly. The localization of TDP-43 G287S also shifted back towards the cytoplasm (Cyt/Nuc ratio of ~0.5 to ~0.8), but the magnitude of the shift was again smaller than that of TDP-43 WT. These data suggest that the nucleocytoplasmic transport process of TDP-43 G287S is less sensitive than that of TDP-43 WT to the changes in development and cytosolic calcium. These additional results are now incorporated into the revised manuscript.

Results

“Taken together, these results support the notion that expression of TBPH/TDP-43 in a location different from its original cell-type specific location is toxic, whether it is a mis-localization from the nucleus to the cytoplasm as observed in *Drosophila* retina (Figure 6—figure supplement 3A-C) or from the cytoplasm to the nucleus as in C4da neurons”

“Next, we examined the subcellular localization of ALS-linked mutant TDP-43 G287S (Voigt et al., 2010) and tested whether developmental process or calcium can alter its localization. As expected, the localization of TDP-43 WT significantly changed from the cytoplasm in larvae to the nucleus in pupae (Figure 2—figure supplement 2A and B). […] These data suggest that TDP-43 G287S is less sensitive than TDP-43 WT to the changes in development and cytosolic calcium in mediating nucleocytoplasmic transport”.